# Blockchain and Secure Element, a Hybrid Approach for Secure Energy Smart Meter Gateways

**DOI:** 10.3390/s22249664

**Published:** 2022-12-09

**Authors:** Carine Zakaret, Nikolaos Peladarinos, Vasileios Cheimaras, Efthymios Tserepas, Panagiotis Papageorgas, Michel Aillerie, Dimitrios Piromalis, Kyriakos Agavanakis

**Affiliations:** 1Laboratoire Matériaux Optiques, Photonique et Systèmes (LMOPS), Université de Lorraine, CentraleSupélec, F-57000 Metz, France; 2Department of Electrical and Electronics Engineering, University of West Attica, 122 44 Athens, Greece

**Keywords:** blockchain, secure element, IoT, P2P energy trading, LPWAN, LoRa

## Abstract

This paper presents a new hybrid approach that is suitable for application to energy smart meter gateways, based on combining both blockchain and Secure Element (SE) technologies serving the roles of a secure distributed data storage system and an essential component for building a “root of trust” in IoT platforms simultaneously. Blockchain technology alone may not completely secure a transaction because it only guarantees data immutability, while in most cases, the data has to be also secured at the point of generation. The proposed combinational approach aims to build a robust root of trust by introducing the SE, which will provide IoT devices with trusted computed resources. The feasibility of the proposed method is validated by testing three different implementation scenarios, using different Secure Element systems (SES) combined with blockchain and LPWAN communication technologies to encrypt, transmit, and save data. This hybrid approach aids in overcoming the obstructions of using any one technology alone, and its use is demonstrated with a case study for an Energy Smart Metering gateway that enables the implementation of a local Peer to Peer energy trading scheme that is end-to-end secure and decentralized.

## 1. Introduction

While renewables are rapidly growing as part of the energy revolution, energy systems structure is becoming progressively decentralized. Once Power Grids (PG) evolve into Smart Grids (SG) while utilizing smart meters (SM) for monitoring and observing various metrics, they prove to be most beneficial in providing major benefits to power utility providers as well as consumers. The SG can assist utilities in reducing expenses and generally electricity costs, improving dependability and transparency, as well as streamlining procedures, while SM techniques promote renewable energy and dynamic pricing for consumers [1]. Generally, the implementation of smart meters (SM), while contributing to the monitoring of a variety of metrics, leads to a more reliable flow of power and provides efficient viewing of information based on power consumption levels [2,3].

In order to accurately meet load demand while minimizing costs, energy retailers invest in collecting data about users’ energy consumption.

As stated in [1], though SMs prove to have an outstanding role in such SG networks, in which the SMs’ sensitivity to the power metrics (voltage and current) is reflected by the fluctuation of these metrics, these networks seem to be the most vulnerable applications that need to be accurately monitored. The increased flow of sensitive information for SMs provokes, against all odds, the possibility of invasion of users’ and power providers’ privacy. Once energy consumption is highly personal, any consumer’s sensitive private information may be misplaced or misused, so measures are taken to increase scalable privacy in SMs without sacrificing the efficiency and cost of implementation. Any undesired data loss may lead to catastrophic situations for power supply companies as well as for individual households. Furthermore, SMs are prone to undesired attacks that falsify the reports delivered to decision-makers leading to catastrophic situations on both the level of human lives and utilities [4].

As SM-based electric power systems are used more often, cyber security risk issues inevitably rise, which raises the need for security methods, therefore leading to crucial security measures regarding SM networks [5]. All previously addressed issues seek a solution that may be derived from blockchain technology. Although primarily designed to enhance decentralized transactions by removing the central authorities from transactional processes, it may also be defined as a distributed ledger technology (DLT) or Internet of Value that securely stores and shares digital transactions without the centralization of management [5,6,7,8]. This structure allows for the automated execution of smart contracts in peer-to-peer trading platforms. Blockchains, on the whole, could be addressed as a global database where multiple users are granted the privilege to modify the ledger and automatically update those modifications by making multiple chained copies of all, making use of the decentralized, multiple-authority management systems structure. Any changes must be approved by the users through consensus mechanisms, which makes this network transparent, secure, and “trustless”.

According to [6] regarding cases of blockchain-related activities in the electricity industry, the blockchain is considered to upstand the highest potential applied to primary activities, both for grid needs and trading, thus urging companies involved in blockchain development for the energy sector to combine various applications and functionalities that facilitate a smart environment. Contrary to centralized energy systems, decentralized systems that involve vast numbers of actors who generate bidirectional electricity flows, consumers might also produce energy and create surpluses at certain times of the day, store energy, and thereby actively participate in grid flexibility through demand-response programs, etc. These opportunities demand increased operational complexity in the transfer of value and information, as well as an effective interface with the physical infrastructure that Blockchain may succeed in supporting as such.

Blockchain-enabled distributed energy resources, like solar panels, photovoltaic (PV) systems, and microgrids that transform consumers into prosumers by enabling active users to sell their excess energy to the grid, are prospering. In this context, the new players, with a low capacity of energy production while unable to participate in the large-scale energy market, could organize Peer to Peer (P2P) local energy trading schemes and participate in virtual power plants without the intervention of any authority or third party.

However, to deliver an effectively secure P2P energy trading, Local Energy Markets (LEMs) require a method for implementing original and secure information and communication technology that Blockchain combined with Secure Elements (SE) and modern Smart Meter Gateways (SMGW) could enhance even further enabling existing energy systems to become more decentralized, secure, smart, and interconnected.

Secure mechanisms are needed that can be implemented at the device level before the data leave the device. These mechanisms should be light and can be implemented in real-time [7]. The growing complexity of the existing infrastructure’s balance at the same time with data security requirements, which are related to the electricity exchange and billing, needs digitization. An intelligent smart metering system and a Smart Gateway (SMGW), which is placed as a relay between Smart Meters (SM), Consumers, and External Market Participants (EMP), can provide a secure way of making this desirable digitization feasible.

This work aims to examine and test different scenarios implementing blockchain and secure element-based SMGW as an enabling technology for energy digitization.

The paper is organized as follows: Section 2 points out the state of the art, Section 3 describes the methodology followed, Section 4 describes the different case studies and present the different results, and Section 5 provides the conclusions.

## 2. Related Work

Today, more than ever, the upward trend of Information Technologies (IT) decentralization over the last decade, along with data processing near the “edge” of the network (Edge Computing), seems to be approaching those days of complete independence from a single central system that collects, controls and ultimately consumes data. Devices (or “Things”) that make up the modern Internet of Things utilize, adopt and integrate architectures, features and capabilities (i.e., Raspberry Pi [9], ESP32 [10], Single/Multi-Core ARM [11] or RISC-V-based [12] Microcontroller Units—MCUs, Microprocessor Units—MPUs, etc.), which constantly seem to approach those of much more powerful systems. Therefore, intense global research activity is justified in the fields of Decentralized and Distributed Systems under the prism of “Constrained Resources Computing Systems” which are usually found as clients—at the lowest level—in IoT environments.

Computing capabilities and data storage tend to move closer to the data generation site, with the ultimate goal of improving response times and saving bandwidth. Edge computing and the speed at which it is evolving offer an extremely important contribution to the overall effort of decentralizing these networks. Such a network will ideally consist of Heterogeneous systems, which may not be already registered in the network but will need to be quickly connected to it securely.

In this cutting-edge period, the boundary between the physical world and the digital world is reducing with the growth of information and communications technologies (ICT). As more and more things are getting digitized, numerous conventional notions from the physical world are being reconsidered. This reevaluation is redefining these conventional ideas, often making them more comprehensive. Notions like signatures, cryptocurrencies, fingerprinting, and transactions are examples of them.

The Automated transfer/transaction of data is one of those traditional concepts that has been redefined completely today. While conventional distributed databases resolve the problem of data integrity, the complicated problem however related to transaction security is still difficult to address. The Introduction of Blockchain (BC) is a new paradigm that has the potential to overcome security and trust challenges for IoT platforms, as a distributed database partially solves the issue.

According to [13], Blockchain is meant to be a transactional database technology evolving to a decentralized way to manage validation and tamper-resistant transactions with consistency across a significant number of members, addressed as nodes. Blockchain can be classified as a type of distributed ledger technology that provides confidence to the user that information, such as certificates, that is archived is not tampered with, thus decreasing transactional obscurity, insecure states, and dubiousness by supplying complete exposure of transactions and the supplementation of homogenous and verified data across all members in the network. By utilizing blockchain technology, trust in transactions is established by a proof-of-work protocol that abolishes any third-party involvement so as to verify and record transactions while avoiding any dependence on third parties for the security of all transactions and assets.

Furthermore, the integration of proof-of-work according to the decentralization aspect of blockchain technology aids computing in resolving various problematic mathematical issues as well as providing a proof-of-work as a well-recognized system for consensus that is currently deployed to reconcile millions of decentralized nodes.

Blockchain alone by itself is not able to completely secure a transaction because it only guarantees data immutability, while in most cases, the data has to be secured at the point of generation. Furthermore, due to its significant overhead, blockchain is not capable of penetrating lower levels in a system. Therefore, to fill this gap, we propose the use of Secure Elements (SE) to build a “root of trust”, and to give IoT devices trusted computed resources to generate a cryptographic signature, following the secure-by-design model. These two technologies, Blockchain and Secure Elements combined, help to overcome the obstructions of using any one technology alone and are used as a base of our suggested end-to-end secure and decentralized IoT system that we will apply and test in an Energy Smart Metering gateway to enable local P2P energy trading.

Taking into consideration the following studies, supported by [7], Blockchain technology for SG applications seems to be still in the research phase. Consequently, the SE solution combined with blockchain mechanisms could be a promising solution to be justified for providing secure mechanisms for SM that can be implemented at the device level prior to data leaving the device itself. These mechanisms may be light and can be implemented in real-time, enhancing computation efficiency, scalability security, latency issues, and integration on the whole [13]. Initially, the concept of secure elements was introduced in [14], and the author provides a generic overview of their features, serving as a starting point to work with secure elements. In [15], the author presents a demonstration of a low-cost, low-power object based on an Arduino board, whose security is enforced by a secure element. In this scenario, a Java Card secure element was used to open a TLS session for device authentication. In the paper [16], some trust issues for blockchain transactions are introduced. Such transactions rely on Elliptic Curve Digital Signature Algorithm (ECDSA) signature based on 32 bytes secret keys. Because these keys can be stolen or hacked, the use of a java card is suggested for secure elements to prevent these risks. This paper focuses on transaction security on the user side. From that perspective, key storage and signature trusted computing are critical topics. In [17], the author presents a new direction for solving the Internet of Things security and trust issues. In this paper, the secure element was used to store the private key and to sign the transaction. We have even integrated an RPi-based controller, which includes a full internet connection, to connect the end node to the blockchain.

### 2.1. Smart Grid and Advanced Metering Infrastructure

A Smart Grid is a new generation of the conventional electricity infrastructure that is to be a solution to improve the electrical energy system not only by integrating Renewable Energy Resources (RES) but also Distributed Generation (DG) and Distributed Storage (DS). Smart Grid targets to solve different existing problems in power delivery, answer the climate changes, improve energy efficiency and localization of energy exchanges and open a new direction in electricity markets (like Peer-to-Peer P2P energy trading). A direct implication of the Smart Grid is to have an electric model that is capable of managing different generation and storage devices in an efficient and decentralized way by deploying an Advanced Metering Infrastructure (AMI) [18]. Advanced metering infrastructure (AMI) is a system that meters, collects, controls, and evaluates the energy consumption at the request site and transmits this information to utilities, grid operators, and customers. An AMI system, as presented in Figure 1, consists of the below-listed components.

Meter: is responsible for energy consumption and generation of real-time metering and transmitting information.Display: is responsible for receiving and processing information, and it shows the power usage analysis.Energy manager: it receives Real-time Transport Protocol/Time-of-use (RTP/TOU) data, or it is preconfigured to conditionally produce signals to control the devices and the condition of the equipment.Smart meter gateway: it is used to exchange information and signals with the higher levels of the hierarchy external and links the components within the AMI architecture.Cloud databank: this is used to record and store the historical power information that can be accessed by externals and customers.

AMI structure’s complexity and simplicity depend on data security requirements, configuration, and connection infrastructure. A different agent can benefit from the AMI like:Customers: by visualizing and presenting the power utilization, these customers may achieve power and price savings.Utilities: saving human resources to collect and record data, proposing new value-added services, and improving the prognosis of demand.System Operators: it offers operators a better knowledge of the grid status and conditions.

The AMI system has been widely deployed and implemented in Europe, the USA, and other countries. Within the restrictions that limit the spread of the AMI, we can name the technical problems, the information security, and standardization and the investment [19].

### 2.2. IoE German’s Intelligent Metering System

IoE architecture refers to communication infrastructure and addresses both the energy and data exchange between different resources and loads, such as sustainable energy sources, distributed energy storage, and residential and modern buyers [20,21]. These communication measures are monitored and verified through the Internet. The overall guideline is that energy and data streams are guided from the sources to the destination, like data streams on the Internet. The Internet of Things (IoT) is seen as the technology that can be used to facilitate two-way communication.

The IoE poses a positive impact and brings important benefits at the national level, which may enable countries to control electricity demand, prevent outages and enable P2P energy trading in the future. Many nations, specifically Germany, have perceived the potential of this disruptive technology and have begun thinking about the best deployment plans.

By giving great importance to standardization and security, the German iMSys, which is considered to be the German version of an advanced meter infrastructure (AMI), is based on two main components: the Smart Meters (SM) and the Smart Meter Gateways (SMGWs), where the merging of both introduce the Intelligent Metering System (Intelligentes Messsystem) [22,23]. The SM by itself is an advanced digital energy meter capable of providing near real-time energy measurements, and it is equipped with a display to visualize the power value at various time intervals. However, the SMGW is the communication component, which is placed as a relay between Smart Meters (SM), Consumers, and External Market Participants (EMP) to provide a secure way of making this desirable digitization feasible. It has two main roles: first, it communicates SM’s energy measures to EMP, and second, it allows the EMPs to send commands back to the local control boxes to adjust the load. External Market Participants are divided into two categories, active and passive, where passive EMPs receive only energy metering measurements or derived information, and active EMPs can transmit commands to local control boxes [7]. An SMGW plays the role of a relay to ensure a communication channel between the devices in the LAN, where we have the Home Area Network (HAN) and the Logical Metrological Network (LMN), and the external entities in the Wide Area Network (WAN). Within the HAN network, generation plants are included (DER), and control boxes are addressed as the Controllable Local Systems (CLS); however, the LMN includes all connected meter units, and finally, the WAN is used for communicating with external entities (like Smart Meter Gateway Administrator” (GWA), and the “External Market Participants” (EMP). It can be considered to be a firewall with smart metering functionalities. It also collects, processes, and stores metering data and provides access to this data only to authorized entities. Before transmitting metering information, these data are encrypted and signed by a Security Module implemented at the SMGW level. Usually, SMGW uses cryptographic services provided by Security Module (like a smart card) to secure the data and assets storage. In LAN, this gateway also provides authorized users with an interface to access their relevant data and diagnostic data to the technicians.

Later on, Smart Meters will be mandatory for installations, and referring to Germany’s Metering Point Operation Law (“Messstellenbetriebsgesetz”, MsbG) [22], the deployment of SMGWs continues a strategy of progressive deployment until 2032, making it mandatory at the end for consumers of more than 6000 kWh/year, or users with sustainable feed-in reaching about and exceeding 7 kW peak. However, for users whose consumption remains below these thresholds, the deployment of SMGW is not mandatory. Theoretically, the obligation to install a smart meter also applies to the use of controllable appliances and installations with a power output of more than 7 kW following the Renewable Energy Sources Act (EEG) and the Act on Combined Heat and Power Generation (KWKG) where the possibility of an automatic Smart Meter that measures transmission to external entities and the receipt of load shifting commands from external entities seem promising. Figure 2 shows the SMGW rollout schedule in Germany.

### 2.3. AMI-Blockchain Architecture

In this section, we focus on the AMI-Blockchain architecture, which is used on smart meters to measure and record the energy usage data of a smart grid. It is a well-known fact that smart meters are connected to networks worldwide to make them more resilient and efficient [17]. Centralized cloud-based architectures are usually preferred to build IoT networks and validate the data for the sensing devices and the end-users. For the validation procedure, we often need a third-party service provider [24]. While the number of connected devices increases, network cyber-attacks against power grids, are also increasing [25]. Network cyber-attacks, where a malicious actor targets the power grid and successfully penetrates security, are happening very often. To avoid cyber-attacks and improve security frameworks, a lot of work and reports have been conducted by computer science to implement smart meters in production [26]. Smart grids do not have an accurate definition, and we can describe them as a two-way delivery of energy, information that is transparent, seamless, and instantaneous, and enables the electricity industry to manage energy delivery and transmission and to empower consumers to take more control over energy decisions [27]. They connect heterogeneous variable components like renewable and non-renewable energy sources, controllers, and intelligent sensors with different functionality and requirements [28]. Smart-grid networks, which are also known as AMI, have changed the electricity grid by adding smart meters [29]. Blockchain technology has transformed smart-grid applications by increasing decentralization and improving trust. Therefore, this technology, along with Blockchain, is used for smart-grid energy management applications. Despite its rapid growth, its advantages are not too exploited in such applications [7,27]. Thus, the deployment of new hardware and software applications is needed. In general, a new AMI-Blockchain application may involve different smart devices, like smart lights, smart sensors and switches, and other electric devices that monitor and control various parameters on a building and can be operated independently or coordinated by one or more users. To achieve the objective of the AMI-Blockchain application, the interconnectivity of smart devices is essential. Therefore, an AMI-IoT Gateway is responsible for interoperability challenges between smart devices. Users can control the smart devices as long as the service provider provides necessary recommendations for controlling smart devices based on prediction algorithms. The Blockchain network is responsible for the connection between different users and service providers to enhance security in the AMI-Blockchain application. The general architecture of AMI-Blockchain is shown in Figure 3.

The various devices in the building communicate with the Blockchain network through the IoT gateway. The data from the devices are placed into blocks, which are chained together using the Blockchain hashing mechanism. The service provider is responsible for the data analysis and for sending suggestions to the users, but it cannot control the devices in the buildings. To start the Blockchain process, a secure private channel is needed. Before the beginning of a new session, the devices and the gateway generate a new key pair. In this state, before taking part in any session, the corresponding generated key pairs are distributed over a secure private channel. The generated public keys are broadcasted throughout the metering system and the AMI-Blockchain application [29,30]. To use the Blockchain hashing mechanism, we need a device, which is called “Secure Element”.

### 2.4. SG Security Considerations

Over the last decade, information systems have undergone a major transformation from proprietary, isolated systems to open architectures highly interconnected with other corporate networks and the Internet. As well, Critical Infrastructures (CIs) are becoming huge infrastructures with multiple access points. However, CIs have expanded, and communication protocols have been developed with no security in mind. Physical attacks on IoT nodes, especially in the case of interconnected nodes, compromise not only the node but also the network security and are more difficult to identify. This exposes CIs to various physical or cyber threats.

As a result, the evolution of power grid systems has led to a significant improvement, which has brought benefits to the utilities, consumers, and the environment. The integration of Smart Meters, the introduction of wireless protocols, and the adoption of bidirectional communication have raised the need for secure communication to protect data transmission. The AMI, which is part of the smart metering network system, is responsible for the transportation of metering data, including billing data and other information. The smart meter network is composed of a large number of smart meters that are interconnected through different wireless protocols to provide the metering data to the control center.

All the SG structure is largely based on the communication networks and the secure delivery and management of smart meters data. This dependence and the software-oriented management of the underlying network make the SG vulnerable to a wide range of threats. These potential threats or attacks could cause many problems in the SG, ranging from physical damage to blackouts. Moreover, attackers could have access to electricity companies’ and customers’ information. In the United States, CIA’s reports have already revealed that hackers have turned off the power in several cities after breaking into electrical utilities and demanding extortion payments before disrupting the power [31].

The U.S. National Institute of Standards and Technology (NIST) has already produced a report that presents a cyber-security strategy and architecture. The document identifies a set of risks for SG, such as:Vulnerabilities and exposure to attacks or accidental errors.New vulnerabilities are generated by interconnections across networks.Vulnerabilities and weaknesses caused by communications network disruptions by the introduction of malicious software.Increased number of entry points and paths available for potential adversaries to exploit.Threats to the confidentiality and integrity of data caused by interconnected systems may increase the amount of private information exposed.Vulnerabilities introduced by the use of new technologies or potential to compromise the data integrity, e.g., customer privacy breaches as a result of increased data collection.

The main problem is that, while the SG exposes the electrical infrastructure to a software-guided framework managed by computer-based control systems—at the same time, these systems are increasingly connected to open networks, such as the Internet, exposing them to cyber risks.

Due to SGs heterogeneous communication architecture, a challenge is to develop sophisticated and robust security mechanisms that can be easily deployed to protect the management of data and communications between different layers of the SG infrastructure. Threats may be categorized according to specific security goals set for:Electricity companies or grid operators;Customers;Home/business environments.

Considering three overall tiers (i.e., layers): (i) the first tier is related to security management in the smart meters, (ii) the second tier is related to the security management at the concentrators’ level and (iii) the third tier is related to the security management at the operator’s level. Concentrators aim to manage the data coming from many smart meters, creating a set of ‘islands’ (i.e., a subset of interconnecting smart meters based on Spatio-temporal characteristics). Different technologies will be adopted at each tier, focusing on different threat categories. A holistic framework is provided to improve SG infrastructure resilience. It is capable of managing both physical (electrical) and cyber threats that are rapidly detected so that remedial actions are affected. A framework is offered, capable of responding to a dynamic environment like the SG, and provides concrete solutions to deal with any security leakage in the infrastructure by maximizing its performance.

Concerning security measures generally protecting Smart Grids against any potential cyber-security threats, defense strategies and some good practices need to be adopted and integrated at the AMI level, including wireless communication protocols and the architecture in use. In addition, the security of all participating components, which addresses the security of each component, including the SM, must be performed and tested. It covers device conformity, functionality, and interoperability testing.

The most relevant attacks against AMI, their impact on the system, and the recommended published best practices are summarized. There are several best practices released for all aspects of Smart Meter, AMI, and communication networks/technologies that can be concluded from the previous sections, where we have discussed their security gaps and vulnerabilities that can lead to a compromised system. The below table, Table 1 mainly focuses on Smart Meter devices, wireless communication technologies, and communication networks in general [32]. Based on different categories within the AMI, it represents different attacks that we could face and the practices to avoid those types of attacks.

### 2.5. Secure Element Introduction

Usually, a Secure Element (SE) is based on a programmable micro-controller MCU which is tamper-resistant and gives a Trusted Execution Environment (TEE) and Trusted Storage Environment (TSE). In general, SEs are tiny in size (25 mm^2^) and produced to provide security capabilities like digest functions (SHA1, SHA2, MD5, etc.) and cryptographic functions (Elliptic Curve Cryptography (ECC), Rivest-Shamir-Adleman Algorithm (RSA), Advanced Encryption Standard (AES), etc.). To accelerate the execution of these operations, a crypto-processor is integrated within the SE.

SEs restricted computational resources usually include less than 1 MB of ROM (200–500 KB) and less than 15 KB of RAM [33]. According to specs, a SE can save up to 5 applications that are named codelets/applets since they are too small apps executing a particular function.

More than ever, SEs are becoming widely used in different fields like payments for credit and debit cards, telecommunication for SIM cards, IoT [34] for authenticity, identification, and verification, and finally, it was used to provide a root of trust in the blockchain-IoT platforms [35], and it contributes to application in different ways like secure boot, secure messaging, and (D) TLS.

Furthermore, there are two well-known SE categories: SEs that are based on Multos (Multi-OS) [36] and SEs that are based on Java Card. Both Java Card and Multos are secure-element-based Operating systems. Multos SE applications are known as codelets, whereas Java Card SE applications are known as applets. Both SE categories are well known in the commercial context. However, Riddle&Code company has additionally introduced their secure element 2.0, which is part of their product range “built for Blockchain” that enables secure storage of the digital identity (which is the private key) on any given device via hardware and software combination. Riddle&Code secure element 2.0 was considered in our testing and simulation case study.

Since Multos SE design is founded on the security by design concept, it requires a specific programming certificate known as Application Load Certificate (ALC). However, for a Java Card-based SE, the same requirement is not valid, as the installation and the update of the applets require a key. The usage of such a key will provide an extra security layer to forbid any unauthorized programmer from compromising the security of the SE by downloading any malicious code on the SE. Since the symmetric key rule is not applicable with Multos SE, as is the case for Java Card-based SE, it offers the capability to be programmed remotely even in an unsafe environment. On the other side, the Riddle&Code SE 2.0 cannot be programmed remotely, but only on the spot, and once configured, no new changes can be made.

Since SEs are a significant category of security applications, they are certified under the Common Criteria-Evaluation Assurance Level (CC-EAL). EAL1 is the lowest certification level, whereas EAL7 is the highest. However, under the EAL7+ level, which is higher than EAL7, few SEs exist.

SEs utilizations mode are either standalone secure MCU or used in combination with a complex system like Hardware Security Module (HSM). Based on the security level and whether the SE supports Public-Key Cryptographic Standards PKCS or not, the prices change greatly. The majority of secure elements are compatible with the Application Protocol Data Unit (APDU) over serial or NFC interface. However, a few secure elements do support the Serial Peripheral Interface (SPI) or Inter-Integrated Circuit (I2C).

In summary, the most important SE characteristics are:Trusted Execution Environment TEE;Presence of a cryptoprocessor and crypto storage;Memory limitations;I2C and SPI support;CC-EAL certification.

## 3. Materials and Methods

The power industry has always been simple. For years, it was based on centralized systems of generation, storage, and utilization with power suppliers and big intermediate traders in the middle, obtaining significant (and often unregulated) margins over the end nodes’ financial benefit [AIP18]. However, the situation is notably changing. The overall technology uprising and the switch to renewables coming from different decentralized resources have shifted the energy balance from centralized energy suppliers to a large number of active prosumers. Prosumers centricity has introduced new challenges to all active members in the energy network. Which can be presented summarized below:Real-time billing.New roles are given to the consumers.Providing energy supply stability and sustainability at both operational and economic layers.Response to consumer’s demands.

Decentralization, which is required to face all these new challenges, needs trust and security. The implementation of a trusted smart meter gateway (presented in the following section) at the prosumer’s premises is the solution to provide a trusted and secure system to answer the above challenges.

Electricity networks worldwide were initially built for one-way power flow, with flat money flowing in reverse. Producers were expected to be larger than consumers, but since prosumers’ potential growth, many challenges are currently under research. Reconstructing power networks is a difficult challenge to tackle but reengineering the electricity market is an additional challenge. With the growth of ICT applications, the connectivity between the physical world (like machines and devices) and the human world has become much greater, a fact that introduces the concept of energy digitalization. Energy digitalization is changing the way by which energy is generated, distributed, used, and sold. The introduction of IoT and blockchain technologies to the energy sector enable energy generation and consumption tokenization and give consumers the possibility to benefit from the usage of green energy.

Blockchain will most probably disrupt the energy sector in the following individual areas [16]:Finance enhancement within distributed energy sources and battery storage.Solution A solution to the split-incentive problem with multi-owner properties.Optimized utilization and increased value of network assets.Affordable and easier access to modern energy markets even for entry-level prosumers.Enablement of a decentralized platform for generating and distributing power.

According to [37], there are seven different research domains regarding blockchain and energy:The decentralized energy marketsMicrogrid and Smart gridEnergy internetSmart contractPeer-to-peerRenewable energyElectric vehicle

### 3.1. Local Energy Market

In the context of P2P local energy trading, prosumers can sell the extra energy produced by their RES to their neighborhood, increasing the operational stability and the financial sustainability of the local market. However, such a system requires a decentralized, trusted, and secure infrastructure to avoid any data alteration between the point of generation and the destination.

In our use case, we need to exchange data between smart meters and a smart gateway and then upload them to a blockchain network. While generating, signing, and sending transactions from constrained resource devices, like a microcontroller-based embedded device plugged into our scenario’s smart meter and other smart meters also (e.g., for other manufacturers—other countries’ standards), are tasks that need sophisticated engineering. Other architectures can also be used to allow the Smart Meter to send measurements to a blockchain network: In any device capable of running blockchain, required cryptography algorithms can directly utilize the pros of smart contracts, thus removing the need for centralized gateways and points of failure.

The trusted energy smart meter gateway is the foundation of a decentralized energy marketplace. It represents the interface between the physical world (PV, smart meter) and the energy marketplace. It enables energy digitalization. Such a gateway connected to an SM enables the implementation of a trusted and secure energy trading system and gives small producers the capability to trade energy and exchange values.

This gateway is equipped with a cryptographic component. This component provides the trusted gateway with the ability to assign a unique identity to the SM and certify it to the blockchain. After this first attestation, the component generates a digital twin to the SM and provides the gateway with an account-agnostic agreement capability. Each fraction of additional power generated on this SM is recorded, signed, and published to the blockchain, ensuring the confidentiality of the published data integrity.

These trusted gateway functionalities can be summarized below:Unique digital identity assignment to the SM to be a certified blockchain client.Enriches the SM with transactional capabilities.The integrity guarantees the connectivity between the gateway and the SM due to the combined use of the cryptographic device.

To meet these requirements, we apply our blockchain-SE-based combinational approach to suggest a new open-source trusted energy smart meter gateway that will be a node in the blockchain. This combination will enhance the system transparency, integrity, root of trust, and non-repudiation by using the SE along with total decentralization im- mutability through the blockchain. In this context, the same energy metering scenario has been designed, implemented, simulated, and presented by taking energy metering data samples every 15 min by using three different SE and blockchain combinations, as presented below:The first one makes use of the BigchainDB blockchain, which is a scalable open-source database that supports multisignature transactions and smart contracts and a secure element developed by Riddle&Code company.The second uses the Helium Blockchain and a secure element developed by Multos to encrypt the payload before sending it to Helium. Then the application server that will receive the data will decrypt the data with the Multos by entering the same key that the node will have. In this way, the encryption key is not shared in the air.The third one makes use of a Blockchain on a SIM implementation, combining IoT connectivity services commercially feasible for low bandwidth IoT applications, such as smart meters provided by 1NCE and data security provided by Ubirch. The latter makes use of hashed, signed, and chained representations of data and is based on a globally unique, patented, an open-source cryptographic protocol for generating trustworthy data streams as traceable identities and documenting the consumption of each individual device [1NCE, Ubirch, Cologne, Germany].

Further, through our simulation, we have successfully demonstrated and tested the realization of according approaches, and we have presented an open-source secure, trusted SMGW to enable local P2P energy trading.

Security consideration, being, in general, a multi-domain, complex, and constantly evolving topic, cannot be exhausted with the comparative implementations presented hereafter in this paper. However, their purpose is to provide a proof-of-concept that will reveal the advantages and disadvantages of each approach.

### 3.2. Case Study I: End-to-End Implementation Architecture

The first use case comprises a crypto wallet implemented in a gateway, as may be seen in Figure 4b which is the middle device between the smart meter and the local market. This trusted gateway is composed of a Secure Element (composed of a Crypto Accelerator Microchip 608A and Crypto Storage Microchip ATAES132A) and RPi.

#### 3.2.1. Experimental Setup and Tools Used

##### BigchainDB Installation

BigchainDB is associated with some specific terminology, like BigchainDB client, node, and network. For the simulation test scenario, we have built our private BigchainDB 2.2.2, which is composed of at least four nodes. Each BigchainDB node, as shown in Figure 4, is composed of the following:MongoDB;BigchainDB server;Tendermint.

In BigchainDB version 2.0.0 and later, each node has its own isolated local MongoDB database. Inter-node communications are performed using Tendermint protocols, not MongoDB protocols, as illustrated in Figure 5 below. If a node’s local MongoDB database gets compromised, none of the other MongoDB databases (in the other nodes) will be affected.

Our scenario implementation is based on using BigchainDB as a database that is characterized by blockchain features. It has high throughput, low latency, and high performance; it is decentralized by design and has built-in asset support. We have used BigchainDB version 2.2.2 using pip3 and docker for installation. BigchainDB server, whose main running process is seen in (Figure 6) has to be installed from the GitHub repository Python library and packages and execute docker functions or directly by using Python pip3.

The data structure is the most specific thing to check and understand about BigchainDB. Contrary to how data is structured in conventional SQL databases (like a table) and other non-SQL databases (like JSON), BigchainDB data is represented as assets. Any kind of data, either physical or object, is considered an asset. Once the node is up and running, a client can connect to the MongoDB either via BigchainDB API to localhost:9984 presented in (Figure 7) or via Shell. However, the user-friendly way is via Compass, which is a free GUI interface and can be used as a tool to interact with your MongoDB. Compass is an interactive free tool to query, optimize, and analyze our MongoDB data.

#### 3.2.2. BigchainDB Node Environment Setup

We used Linux (Ubuntu 18.04 and amd64 architecture) operating system for a simple and fast setup, and during the setup and simulation phase, we used PYTHON 3.6.9 as the programming language.

Figure 8 depicts the versions of the different components that are used to set up our testing environment.

#### 3.2.3. Secure Element Setup

The Riddle&Code Secure Element was developed with dual targets, one to extend the Arduino UNO R3 pin and two to extend all Raspberry Pi pin-compliant boards. Female pins are used to connect the secure element to the Raspberry Pi; however, male connectors pins can be connected to Arduino UNO. Figure 9 shows how the secure element is connected to our RPi.

Once connected and configured to the secure element on RPi, there is also a pre-required environment that needs to be prepared. First, to enable the communication between the SE and RPi via I2C should be configured like “Activate i2c via: raspi-config”, in addition to the installation of some libraries like “libcryptoauth-0.2”.

The I2C bus permits different devices to be connected to our RPi, each device with a unique address that can often be set by changing jumper settings on the module. It is important to be able to identify which device is connected to your Pi just to make sure that everything is working properly.

The Secure Element is composed of a Crypto Accelerator Microchip 608A and Crypto Storage Microchip ATAES132A. At the setup phase, the SE will generate a pair of keys to identify the SM to the blockchain (those keys are generated by the SE that is capable of generating up to 15 pairs of keys per slot, though once the slot is locked, no other pairs of keys may be generated for the locked slot).

#### 3.2.4. Combinational Approach Testing Results

The system is up and running and ready for testing after finalizing the hardware and software setup. The collected data from the sensor will be forwarded via the RPi to the secure element. In its turn, the SE will encrypt and sign the data with a pair of generated keys at slot 0.

The secure element will return the data encrypted and signed to the RPi, which in its turn will transmit this data to BigchainDB. Below is the secure element output (Figure 10):

Transaction in BigchainDB: in BigchainDB, transactions are usually used to register, issue, create and transfer data (called assets). An asset can be any physical object like a bike or car or a digital asset like a customer object. In transactions, assets are immutable, and each asset can have metadata that is mutable and updated in the next transactions. Transactions have an input and output and are the basic type of records stored by BigchainDB. There are two types of transactions: CREATE and TRANSFER.

CREATE Transactions: the create transaction function is used, as already mentioned, to create, issue, and register an asset being the history of an asset in BigchainDB. A create transaction has one or more outputs. Every output has an associated number of shares. For example, if 50 oak trees are associated with an asset, one output may have 35 oak trees for a group of owners, and the second output may have 15 oak trees for the second group of owners. Every output also has an associated condition: conditions must be fulfilled by transfer transaction to transfer the output. BigchainDB has a different variety of conditions.

In Figure 11, we see a diagram that presents a BigchainDB CREATE transaction that has one output which is Pam, who owns and controls three shares of the assets, and there are no other shares since there are no more outputs.

A TRANSFER transaction: transfers ownership of an asset by providing an input that meets the conditions of an earlier transaction’s outputs. With a transfer transaction, you can have one or more outputs just like the create transaction. Moreover, the total number of shares coming in on the inputs must be equal to the number of shares going out on the outputs Figure 12.

Let us consider the creation and transfer of a digital asset between two neighbors (A and B). This asset represents the extra available energy metering data. We will suppose that the extra energy data belongs to neighbor A and will be transferred to neighbor B, knowing that this data is the output of the secure element presented in Figure 13.

To create a transaction, neighbor A has to define the asset and then create a transaction. Asset definition: BigchainDB users are identified by public/private key pairs. The private key is used to sign transactions, while the public key is used to verify that a signed transaction was truly signed by the one who claims to be the signer. Now neighbor A is ready to create the asset, and the transaction has to be fulfilled by signing it with a neighbor A private key. Once the asset is created and signed by the owner, it can be transferred to the BigchainDB node. Usually, after a few seconds of sending the transaction to the BigchainDB node, it is important to verify that the transaction was successfully sent, validated, and included in a block. If the transaction succeeds, this will return the block height containing the transaction. However, if it fails to do so, then no block will be generated and will obtain None as a return. There are different reasons behind the None; for example, it could be related to the validity of the transaction or to delay, and the transaction is still in the queue. Usually, an exception is raised in case of an invalid transaction. After running the creation code, block creation starts as in Figure 14.

To check the whole block, we can use the block height to retrieve the block itself or also use Mongo compact to interact easily with mongo DB, as presented in Figure 15.

After neighbor A successfully created the asset and submitted it to the blockchain, it decided to transfer asset ownership to neighbor B. Neighbor A retrieved the transaction id “creation_tx = bdb.transactions.retrieve(txid)”. To prepare for the transfer transaction, neighbor A has to know the asset id. Once the id is retrieved neighbor, A can prepare the transfer transaction, fulfill it, and finally, sends it to the connected BigchainDB node. Now, neighbor B is the owner of the asset, and neighbor A is the former owner. Now neighbor B has access and is the owner of the asset and is capable of decrypting the data with his secure element.

The feasibility of combining blockchain and secure elements was investigated and argued on its benefits and demonstrated with experimental implementation. The proposed prototype is tested using BigchainDB and Riddle&Code secure element. The adoption of the above scenario allowed secure data transmission between two clients and presented a secure prototype for the IoT platform. The developed code permits the creation and transmission of energy data between two neighbors.

### 3.3. Case Study II: Using Secure Element and Helium Blockchain

For the needs of connecting IoT devices to the internet, special wireless LPWAN networks were created and set out to accomplish energy efficiency, scalability, and coverage, at low data rates unlikely, for example, Wi-Fi or 4G technologies [38,39].

The LoRa network is one of the most widely adopted LPWAN standards internationally. LoRa technology enables the long-range communication link and belongs to the physical layer (PHY) of LoRaWAN, which defines the communication protocol and network architecture and belongs to the media access control (MAC) layer [40].

Combining LPWAN with the right blockchain can provide a viable solution to the IoT system. Using Blockchain offers security and trust in an IoT system as each new block is linked to all its previous blocks in a cryptographic chain in such a way that it is almost impossible to hack [41]. Blockchain is a good security solution and enhances data integrity, but in some cases, its use can be unprofitable.

#### 3.3.1. Helium Blockchain, a Public LoRaWAN Network

The Helium Blockchain is the world’s largest public decentralized LoRaWAN network, a blockchain that allows devices anywhere in the world to be wirelessly connected to the internet and geo-located [42]. It is a good choice for interconnecting IoT devices for the following reasons:LoRa nodes can connect to the Helium blockchain through the available coverage hotspot;Helium ensures that all communications on the blockchain are end-to-end encrypted, making it more suitable for sensitive information;It has low transaction fees. When sending the data received from the sensor for every 24 bytes is $0.00001;There is great network coverage from hotspots where we can connect our IoT devices. This leads to a reduction in hardware costs since it does not require us to have a hotspot.

In the link https://explorer.helium.com/hotspots (accessed on 12 August 2022), we can see the active hotspots and coverage worldwide.

Figure 16 shows the hotspot coverage for the region of Attica in Greece.

The space where the NwkSKey and AppSKey keys are stored on the nodes is likely not secure, and if a malicious user manages to “steal” the specific keys, he can manage to decrypt the payload. To address this particular vulnerability, we suggest using the Secure Element: Multos Trust Core to encrypt the payload before sending it to Helium with AES-CBC encryption. The payload will not be decryptable even if the malicious user has the NwkSkey and AppSKey.

The implementation of SMGW is given in Figure 17 and has the name STRESQLab Smart Metering System. The STRESQLab Smart Metering System consists of two basic levels, the STRESQLab-SMGW and the STRESQLab Smart Metering Server. STRESQLab- SMGW will send the data securely to the STRESQLab Smart Metering Server. The data from the Helium Blockchain will be received on the STRESQLab Smart Metering Server through the MQTT Broker, where the payload will be decrypted using another Multos Trust Core where the same key that STRESQLab-SMGW has will be entered. With this place, the AES-CBC encryption key will not be shared over the air.

#### 3.3.2. STRESQLab-SMGW

The smart energy meter gateway (STRESQLab-SMGW) is given in Figure 18 and consists of the following:Embedded system: Raspberry pi zero 2wOS: Raspbian GNU/Linux 11 (bullseye)Secure element: Multos Trust CoreLoRa node hat: Pi Supply/pHAT PIS-1128.Sensor: RPICT3V1Raspberry gpio expansion: (RaidSonic Icy Box IB-RPA101)

Raspberry pi zero 2w was chosen because it is low cost and easy to connect to the secure element: Multos Trust Core. LoRa node hat: Pi Supply/pHAT PIS-1128 is the only hat that works on pi zero, and it works very satisfactorily. With the use of Ppi gpio expansion (RaidSonic Icy Box IB-RPA101), it is more flexible to connect the LoRa node hat, the Multos, and any sensor.

#### 3.3.3. LoRa Node Hat: Pi Supply/pHAT PIS-1128

This LoRa extension connects to the Raspberry pi. It supports all Raspberry models as well as the zero series. The support of the pi zero series gives a great advantage to this module as LoRa nodes can be implemented at a low cost. It uses the RAK811 chip which is based on the Semtech SX1276 and supports various frequencies except for EU (863–870 MHz), such as US, AU, AS, IN, and KR.

The pHAT PIS-1128 communicates with the Raspberry Pi over UART only using a total of 3 GPIO pins for the module [43,44]. Figure 19 shows the GPIO used by the module:

To enable communication between raspberry and the module, it should be set as “Enable Serial via raspi-config”. Then it will be necessary to install the rak811 library via terminal and pip3 python command.

In Table 2, they are given the operating features of the Helium Node.

The location where the NwkSKey and AppSKey keys are stored on the end devices is likely to be insecure, and if a malicious user manages to “steal” the specific keys, they may be able to decrypt the payload. RAK811 probably does not provide a secure environment for storing or using these keys. To address this particular vulnerability, we suggest using Multos to encrypt the payload before sending it to Helium with AES-CBC encryption so that we do not rely on the weak security of the RAK811. The payload will not be decryptable even if the malicious user has the NwkSkey and AppSKey.

#### 3.3.4. Secure Element: Multos Trust Core

MULTOS Trust Core is a high-security embedded microcontroller that provides a hardware security module for smart and connected devices. It offers hardware protection from the principle of trust (Root of Trust (RoT)) as it generates and protects keys and performs cryptography operations within its secure environment. It connects with Raspberry pi and Arduino [45,46].

The Multos Trust Core communicates with Raspberry via I2C. Figure 20 shows the GPIO used by the Multos Trust Core:

To enable communication between the raspberry and the Multos, it should be set as “Activate i2c via raspi-config”. Then it will be necessary to install the i2c driver and PKCS#11 library. In this particular scenario, Multos will be used to encrypt the payload before sending it to Helium Blockchain using AES-CBS encryption. A build for the Multos should be installed where it will include two files, pkcs11.h for using library PKCS#11 to call function aesEncCbc and aesDecCbc to encrypt and decrypt the payload and the file Python.h for building the Python module.

To import the key in the Multos that will be used to encrypt the payload via the terminal we execute the command:

p11keyman -i 32 ED AES_CBC

The same key will be imported in the Multos of STRESQLab Smart Metering Server to decrypt the payload

Using the following command displays the keys entered in the Multos:

p11keyman –l

#### 3.3.5. STRESQLab Smart Metering Server

The STRESQLab Smart Metering Server consists of the following:Embedded system: Raspberry pi4 4GB (OS: Raspbian GNU/Linux 11 (bullseye));Software: Node-Red, Azure IoT Hub, Mosquitto broker, Paho-MQTT, Influx dB, Grafana;Secure element: Multos Trust Core.

The STRESQLab Smart Metering Server executes node-red, and Figure 21 shows the design it follows:

Through the MQTT Broker node, we receive the data from the Helium Blockchain, i.e., the data we have sent from the STRESQLab-SMGW (the node). The JSON node converts between JSON String Objects. The base64 node converts the payload from base64 format to string format. The Node-Red module payload is set to the payload. Payload sets the payload to be just the message that the node has sent with no other information. The payload is then published to the Mosquitto MQTT broker in the data_payload topic via the data_payload node.

Figure 22 shows the encrypted data sent from the node to the helium blockchain. Data is encrypted before being sent to the Helium Blockchain.

Figure 23 shows the encrypted data that the STRESQLab Smart Metering Server receives from the Helium Blockchain. Then using Paho, we subscribe to the data_payload topic, use the code in the script so that Multos can decrypt the payload, and publish the decrypted payload to the decrypt_payload topic.

The decrypt_payload node connects to InfluxDB, and via Grafana, we have data visualization, as shown in Figure 24.

### 3.4. Case Study III: Blockchain on a Sim Offering Combined IoT Connectivity Security in One Solution

The third solution under test makes use of a trust protocol for data packets directly from the sensors’ measurements, whereas stated, the data are sealed and chained with cryptography to become immune to manipulation once stored in a Blockchain.

Such a security solution is provided by 1NCE combined with UBIRCH by adding a blockchain security component to their IoT FlexSIM card, including a private key sealing IoT data (IoT Flex SIM card connectivity by 1NCE) to combine high-quality IoT connectivity with blockchain-based security backed up by Ubirch components. The outcome is a Blockchain on a SIM solution tested in this article where the sensing devices manage their own blockchain identities and become direct actors in the blockchain system, providing a “root of trust” for the sensor data [24].

Via NB-IoT, 2G, 3G, 4G, and LTE-m, 1NCE [47] acts as a Mobile Virtual Network Operator offering data connection, making use of TCP or UDP transport protocol and a cloud-based core network. It provides IoT-grade plastic SIM cards in any factor forms such as nano, micro, and standard Crypto utilizing blockchain capabilities based on UBIRCH [48] nano client, Blockchain applet, and certificate included on the SIM chip. It addresses edge devices, tiny MCUs, and SIM cards securing IoT data at the sensor, checking for integrity of measured and transmitted values if they may be hacked, deleted, or duplicated checking the identity of the sensor as well. In Figure 25 below, the whole concept of the device application software and the SIM application included in the smart IoT device is displayed, engaging the customer’s backend application with the Ubirch trust service.

In addition to the dedicated data channels implementing channel security solutions with TLS or VPN techniques for sending data from sensor measurements, Ubirch uses a nano client as part of firmware in the sensor itself that creates a signature every time a reading is made to be sent to the receiver. Additionally, it creates its Keypair of private and public keys based on the SIM card’s IMSI for cryptographic identity and verifies data through Blockchain anchoring every time a measurement is taken. Utilizing a Trust Service, Micro-certificates are stored in a cloud-based backend and anchored in public blockchains such as Ethereum, Ethereum Classic, and IOTA, thus creating an immutable and irrefutable record of that sensor.

#### 3.4.1. Using UBIRCH Blockchain with 1NCE SIMs

The SIM application client provides signature and chaining services to seal original data generated on embedded devices through the SIM card while original data is stored in a customer database to be able to execute verification requests at a later stage.

The Trust Service, as a cloud-based backend, creates its Merkle-tree structure, aggregating incoming UPPs into larger root- hashes, which get anchored into a blockchain every minute, thus keeping costs down [49]. It ensures the packaging of the hashed data and the signing of the package into the UBIRCH PROTOCOL PACKET (UPP). The anchoring in the blockchain is utilized at the backend, which can also be used to verify already anchored UPPs. No data are stored in UBIRCH.

#### 3.4.2. Utilizing the Hardware Setup with the UBIRCH Testkit

The UBIRCH Testkit hardware that we made use of, as shown in Figure 26, is generally a MicroPython client for the UBIRCH protocol on a SIM, thus programmed and configured with the UBIRCH nano client. It is based on Pycom hardware modules such as the GPy [50], a Pycom triple–bearer MicroPython-enabled microcontroller based on the Espressif ESP32 SoC. It supports Wi-Fi, BLE, and cellular LTE–CAT M1/NB1 connectivity, as well as the sensor board Pysense V2.0 [51] that is a multi-sensor board that comes in the shape of a shield for connecting the GPy to it.

Both boards utilize an enterprise-grade IoT platform for connected Things to test the usage of the UBIRCH protocol, provided the micro python example code is used for the purpose provided by Ubirch via GitHub [52]. Such a setup was used in the present research for evaluating the ’blockchain on a SIM’ application which implements UBIRCH functionality inside a SIM card. The SIM card needs to have the ’SIGNiT’ [53] applet to support cryptographic functionality.

The test kit components, as can be seen in Figure 27, are:1NCE SIM Card with SIGNiT mobile security application;Pycom Gpy;Pycom Pysense;Pycom cellular LTE/NB-IoT antenna attached to the Gpy;micro-SD card.

#### 3.4.3. Initializing and Authentication of the Setup—Unlocking the SIM

On initial startup, the Testkit will load the configuration from the SD card and connect to the NB-IoT network (APN: iot.1nce.net). So, to activate the SIM card, we make use of the UBIRCH backend [54] by claiming the card via registering its international mobile subscriber identity (IMSI). That is an internationally standardized 15-digit unique number to identify a mobile subscriber, as defined in ITU-T Recommendation E.212 [55].

As featured in Figure 28, to enroll our NB-IoT device in the Things console with the IMSI number, our personal login credentials, such as our e-mail and password, are needed. Then the following authentication keys are generated:(UU)ID: 07104301-xxxx-xxxx-xxxx-0000dc7b0efd an ID key of 16 Bytes and;password: 32f86549-xxxx-xxxx-xxxx-xxxxxxxxx as an authentication token.

The Testkit will create an apiConfig text file on the SD card, which contains the unique IMSI of the SIM card as well as the following elements:“keyService”: “https://key.prod.ubirch.com/api/keyService/v1/pubkey/mpack (accessed on 20 July 2022)”;“niomon”: “https://niomon.prod.ubirch.com/ (accessed on 20 July 2022)”;“data”: https://data.prod.ubirch.com/v1/msgPack (accessed on 20 July 2022).

After the initial procedure and authentication keys are fully functional, the device performs a bootstrap with the UBIRCH backend to acquire the SIM card’s PIN via HTTPS. Once the SIM card is unlocked, the device always reads the pin and IMSI from the flash memory and is ready to send signed UBIRCH Protocol Packages (UPPs) to the backend to be anchored to the blockchain.

#### 3.4.4. Connecting and Sending Data

The device was programmed to take measurements once every ten minutes and send a data message to the UBIRCH data service. The data message contains the device UUID, a useful timestamp, and a map of the sensor data for testing purposes as below:“AccPitch”: <accelerator Pitch in [deg]>, “AccRoll”: <accelerator Roll in [deg]>,“AccX”: <acceleration on x-axis in [G]>, “AccY”: <acceleration on y-axis in [G]>,“AccZ”: <acceleration on z-axis in [G]>, “H”: <relative humidity in [%RH]>,“L_blue”: <ambient light levels (violet-blue wavelength) in [lux]>,“L_red”: <ambient light levels (red wavelength) in [lux]>,“P”: <atmospheric pressure in [Pa]>, “T”: <board temperature in [°C]>,“V”: <supply voltage in [V]>

The raw data mentioned above, in JSON format, are initially guided via an independent route toward an open-source PHP SQL database and finally to the open-source observability platform Grafana for aggregating and visualizing data, as in Figure 29.

#### 3.4.5. Sending Hashed Data Program

On the other hand, data is hashed, encapsulated in an Ubirch Protocol Packet, to be anchored in Blockchain as can be seen below:

data = {‘AccZ’: 1.025757, ‘H’: 29.06851, ‘AccPitch’: 2.20808, ‘L_red’: 7,

‘L_blue’: 5, ‘T’: 34.25, ‘V’: 4.536165, ‘AccX’: −0.00390625, ‘P’: 101049.5,

‘AccRoll’: 0.225009, ‘AccY’: −0.03393555} data message [json]: {“data”:{“AccPitch”:“2.01”,”AccRoll”:“0.97”,”AccX”: “−0.02”,“AccY”:“−0.04”

,“AccZ”:“1.03”,“H”:“28.30”,“L_blue”:24,“L_red”:28,“P”:“101107.00”, “T”:“33.56”,“V”:“4.53”}

,“msg_type”:1,“timestamp”:1656767975,“uuid”:”07104301-1892-4020-9043

-0000dc7b0efd”} # creating UPP

UPP: 9623c410071043011892402090430000dc7b0efdc4404dcfaac6395279798......

# 300 characters

data message hash: OEQsbWRnu0qxp8gsm1DFM6nYS2yO2iRmzscZtywuX2Y=

connecting to the NB-IoT network # checking/establishing connection sending...

# sending data

sending...

# sending UPP

time synced # waiting for time sync

close connection # preparing hardware for deepsleep deinit SIMdeinit LTE

>> going into deepsleep for 548 s

To send the readings and utilize cryptographic sequences, Ubirch Protocol is needed to obtain UUID from SIM Card and send a Sign request for the public key that is shown in the console. The response from Ubirch X.509 certificate contains the SIM applet PIN to unlock crypto functionality.

Finally, the device is ready to send signed UBIRCH Protocol Packages (UPPs) to the backend to be anchored to the blockchain [56]. UBIRCH Protocol Package (“UPP”) is generated with the unique hash of the serialized data, UUID retrieved from the SIM card and timestamp, chained to the previous UPP and signed with the SIM card’s private key using the crypto functionality of the SIGNiT applet. The private key is stored in the secure storage of the SIM card and cannot be read by the device. The sealed data hash is then sent to the UBIRCH authentication service (“Niomon”), where it is verified with the SIM card’s public key and anchored to the blockchain [56].

#### 3.4.6. Verifying the Blockchain Anchoring in the UBIRCH Console

We may verify the Data anchored in the blockchain by selecting the “Recent Hashes” tab, as seen in Figure 30A, and then clicking the verification button to be transferred to the verification page to insert the hash of the data message and witness the anchoring of the particular UPP in Ethereum classic and IOTA Blockchain anchors either in graphic mode (Figure 30B) or as a JSON script as can be seen below in the Blockchain.

The most significant part of the Script of Blockchain anchoring is listed below.

“label”: “PUBLIC_CHAIN”,

“properties”: {

“timestamp”: “2022-08-01T17:29:18.484Z”,

“network_info”: “Ethererum Classic Mainnet Network”,

“network_type”: “mainnet”,

“txid”: “0x6c4971391fb9526612f99591e020529cb91a5a9d30ab9c1 xxxxxxxxxxxxxxxxx”,

“hash”: “0x6c4971391fb9526612f99591e020529cb91a5a9d30ab9 cxxxxxxxxxxxxxxxxx”,

“status”: “added”,

“public_chain”: “ETHEREUM-CLASSIC_MAINNET_ETHERERUM_CLASSIC_MAINNET_NETWORK”,

“blockchain”: “ethereum-classic”,

“message”: “f853468843654432a8990a5e9d993a79145xxxxxxxx00 xxxxxx0a201e9d354dffb99926ca1eedb9d99b2daaaa578 ba91a1db2aeab1de1b52d5b61bcxxxxxxxxx”,

“prev_hash”: “f853468843654432a8990a5e9d993a79145xxxxxefc xxxxxxxx0a201e9d354dffb99926ca1eedb9d99b2daaaa578ba91a1db 2aeab1de1b52d5b61bcxxxxxxxxx”,

“created”: “2022-08-01T17:29:17.477Z”

The success of the Data verification process is witnessed in the UBIRCH platform, as may be seen in Figure 30A. Additionally, the user may seek the anchoring procedure in Ethereum classic or IOTAA in a graphic mode in the UBIRCH platform as well as in Figure 30B. Such information proved to be useful giving us proof of Blockchain anchoring in any stage of anchoring hashed data.

#### 3.4.7. Data Usage in Cloud Platforms

For collecting and processing streamed data records in near real-time, we managed to integrate the Data Streamer platform provided by 1NCE into the Kinesis cloud platform provided by AWS. The latter makes use of AWS IAM Trust Relationships to establish a trustworthy data transferring procedure to the cloud (Figure 31).

The setup of AWS integration for our setup was accomplished via the 1NCE Portal. Data usage monitoring is a noticeable fact in smart meter implementations. Data usage is visualized in 1NCE’s portal as well and has been summed up in Table 3, reaching 3.03 MB in a time frame service of 30 h, including both uplink and downlink data packets, overhead data generated by UDP or TCP protocols, and for the test setup, the actual payload claimed to be the user data derived from the sensor’s measurements.

Approximately six measurements are taken per hour, creating 180 UPPs that were sent in 30 h of service, meaning that 126 KB of data were sent approximately every hour. However, the useful data referring to data anchoring in Blockchain are far less, as can be seen in Table 3. The maximum bit rate was set to 1 MB/s while, in practice, it is measured in real-time to be approximately (median value) at 5.2 KB/s.

Accordingly, data usage streams provided to the AWS metrics platform are visualized in the AWS platform itself. The same measures are displayed as can be seen in Table 3, where the Data rate is measured to be about 5 KB/s while rarely reaching a maximum of 13 KB/s.

Anchoring may be a time-consuming procedure, so in Table 4, the latency of the sent data records is provided as measured. In our setup, it is kept as low as a 10 ms average.

On the other hand, it seems that all records have been successfully received, as can be seen in Table 4, i.e., all data usage packets sent in the predefined period of 15 min were successfully forwarded and added to the blockchain.

#### 3.4.8. The Cost of Cloud Platforms Uses

The 1NCE portal platform cost is included in the 1NCE IoT Flat Rate, rated to 12 euros for 10 years of IoT connectivity for each SIM card purchased as a bundle of a minimum of 200 cards. Additionally, using Blockchain anchoring with all features included while operating Ubirch Trust Service, extra fees should be taken into account due to escalating costs on an annual basis.

On the other hand, the flexible AWS platform costs went by a pay-as-you-go pricing scheme regarding the capacity modes to be used. Making use of AWS Kinesis Data stream dashboards and analytics tools proved to be useful, providing a variety of data usage dashboards though costly. For a total of three months of usage and a single SIM, the total costs rounded up as can be seen in Table 5 below.

So, in Table 5, it is made quite explicit that for a single SMGW with one 1NCE—Ubirch SIM Card setup, using Amazon Web Services, expenses add up to €17.12 monthly average. Trying further to make provisions regarding the Data Usage and Cost of a total of 200 SIM Cards, we made use of the online Amazon cost configuration calculator [57]. For such a case study and according to the Configure Amazon Kinesis Data Streams calculator, for 200 SIM Cards, i.e., 200 gateways that use 5.2 KB/s referring to our setup scheme, the following assumptions were made:Baseline number of records is 20 records per minute leading to 0.33 records of data per second.Peak number of records is estimated to be 20 per minute, which is 0.33 records per second.A Buffer needed for growth and to absorb unexpected peaks of data should be 20/100 = 0.2.

A uniquely identified group of data records in a Kinesis data stream is deducted from the Peak number of records, and the buffer called a shard and is calculated to be equal to 2, i.e., Shards per month = 2.
The duration of data retention is one day, i.e., 24 h.

Finally, the pricing calculations lead us to 4.96 GB as a Baseline data volume per month. Total two billable shards × 730 h per month × USD 0.02655 equals USD 38.76, meaning that the Kinesis data stream cost (monthly) is 38.78 USD or 465.36 USD annually for 200 SIM cards based on the previous assumptions regarding data rates in Table 5 (in the discussion division).

We should note that 1NCE—UBIRCH as AWS Services costs are monthly based as well as on an annual basis according to data usage and quantity of SIM cards used. In particular, the 1NCE IoT Flat Rate, including a Blockchain IoT FlexSIM card, is referred to as a Lifetime fee for 10 years of service per SIM, including 500 MB of data and 250 SMS, as well as the Connectivity Management Platform. Additionally, all Ubirch Services following the configuration of UBIRCH Trust Service include the following:Setup of a customer-specific tenant in the UBIRCH Trust Service (cloud), provision of APIs, etc.;Public blockchain connection (Ethereum, Ethereum Classic and/or IOTA);Additional fees should be considered while operating the UBIRCH Trust Service, including:Operations and license fee for 200 SIM cards, incl. 50,000 transactions/SIM;Archiving duration: 12 months and Blockchain anchoring frequency: 1/h;€5.00 p.a. per SIM card.

Usage fees regarding our implementation are quite explicit and clear once we do not opt for an open-source platform.

In Table 6, the overheads of all three setups are displayed.

## 4. Results and Discussion

The feasibility of combining blockchain and a secure element was investigated and argued on its benefits and demonstrated with experimental implementation. Initially, the proposed prototype was tested using BigchainDB and Riddle&Code secure element, allowed for a secure data transmission between two clients through LAN, and presented a secure prototype for the IoT platform. The developed code of this scenario permits the successful creation and transmission of energy data between two neighbors.

Our case study one is a zero-cost solution and a quasi-open-source solution. The Riddle&Code secure element was provided for free by the company for testing. The company dedicates several secure elements annually for new tests. Additionally, for BigchainDB, which is an open-source blockchain, there are two ways to run/use BigchainDB. BigchainDB can be used for private deployments, in which case federation members would need to handle the costs of running a BigchainDB node or use a public deployment of BigchainDB like Interplanetary Database (IPDB) in which case fees are applicable (see Table 7) associated with using the service. The private mode was used in our scenario. In order to run a single BigchainDB node, a single Virtual Machine (VM) running all the components (BigchainDB Server, Tendermint, MongoDB, maybe NGINX, etc.) is only needed. That can be quite a low-cost solution. However, of course, a proper BigchainDB network should have at least four nodes, all run by different entities. That way, one node can fail arbitrarily, and the network will continue to work properly.

The setup of such a test is somehow challenging since you cannot find an updated GitHub setup tutorial of the Riddle&Code secure element, and BigchainDB is still under enhancement and development.

The scenario was tested by sending a data packet every 15 min, as all three scenarios, as stated before in the text, according to European standards. The content refers to metering data that was sent to the SE to be encrypted and signed before being transmitted to BigchainDB. However, the number of transactions that can be created and published depends on the number of bytes. For example, for processing a packet with a size of 765 bytes, one million transactions were processed in about 56 min without any failure, while 99.7% of transactions were finalized within 9.39 s, 95% within 5.14 s, and 68% within 2.85 s. The latency of anchored Data in the Blockchain is displayed in Table 6. The network finalized an average of 298 transactions per second, with a median value of 320 transactions per second.

In the case of Helium, based on LoRa communication protocol, blockchain was used where LoRa nodes can be connected and ensure that all communications on the blockchain are end-to-end encrypted, making it more suitable for sensitive information. There is sufficient coverage of available hotspots that can be used. The Helium blockchain has low fees (24-byte payload = €0.0000103), giving a viable solution for the implemented IoT system. Because Blockchain cannot fully secure a transaction and data must be secure at the point of production, the secure element Multos Trust Core was used at the node, and the payload was encrypted with AES-CBC encryption before being transmitted to the Helium Blockchain providing the system with Root of Trust protection. The STRESQLab Smart Metering Server also used the secure element Multos Trust Core to decrypt the payload from the node. The same key was imported into Multos as the node used, and that way, the key was not shared over the air. As seen in Figure 23, the results were satisfactory. The peak and average data rates (see Table 7) are kept low, as well as the maximum possible data transfer rate of the channel due to the open LoRaWAN protocol restrictions.

Making use of Ubirch/1NCE blockchain on a SIM solution based on NB-IoT communication technology, in conjunction with the EWS platform, we utilized a decentralized computing platform that runs smart contracts, i.e., Ethereum Classic and IOTAA, which is an open, feeless, and scalable distributed ledger, that stores records of ownership of digital assets making use of all security measures that blockchain implementations have to point out. NB-IoT offers low latency (see Table 7), a low Data rate compared to the rate provided by LoRa, and less than that provided by the End-To-End architecture.

Even though such an implementation is not based merely on an open-source scheme, such a solution proved to be versatile, adaptable to the user’s needs, and easily programmable via the well-established full python compiler, the MicroPython, aimed at microcontroller and small embedded systems implementations and cloud-hosted. Once data may be transmitted via a different route than the trusted protocol packets that are transmitted via NB-IoT, indoor coverage is obtained, besides low cost, long battery life, and high connection density, while combining appropriate security standards of blockchain technology in addition to those of mobile radio.

Sending signed transactions every time a measurement is taken right on the sensor’s point while proving ownership of a blockchain address seals data directly at the source, a fact proven to offer security convenience and tamper-proof data transmitting.

Taking into account the cost of use, the total amount of 500 MB of data provided by 1NCE, according to the flat rate policy, before recharging the SIM card, proved to be somehow adequate though restrictive. We suggest that less frequent measurements should be taken and transmitted to the cloud, and the data should be omitted from UPPs to limit data usage and keep costs down.

Making use of AWS Kinesis Data streams dashboards and analytics tools proved to be useful in providing a variety of pieces of information, the outcome of which are summarized in Table 3, Table 4 and Table 7. The use of these tools is to be justified in future work.

The most outstanding performance facts as stated before, for all three case studies/scenarios are displayed in Table 7, as well as fees and costs. Specifically, the terms Μax. The data transfer rate implies the maximum data rate of each data channel, while the Peak Data rate addresses the measured maximum data rate of one IoT node as with one SIM card engaged with 1NCE and Ubirch services in the third case/scenario.

Regarding the costs, we should furthermore state that the costs of the services apply to the AWS Kinesis fees for one node/SIM card and Amazon fees regarding Bigchain database services, while the Total Annual Blockchain services costs refer to the total Annual AWS Kinesis usage in addition to the UBIRCH Trust Service costs based on provisioning for 200 SIM cards. LoRa-Helium proves to be a low-cost solution although posing high latency in the region of seconds in contrast to the other two solutions that pose latency in the region of milliseconds.

All three cases proved to be totally successful in sending data and anchoring the data at the source to the blockchain, taking into account the 15 min of sending period, posing no data loss at all, as stated in the cloud platforms, while providing quality metrics.

Whether using BigchainDB as a database or Helium Blockchain as a Cloud Databank with Multos Trust Core or Blockchain Ethereum, Ethereum Classic, and IOTAA in combination, all three implementations improved the security of data sending as they served simultaneously the roles of a secure distributed data storage system and an essential component for building a “root of trust” in IoT platforms while combining both blockchain and Secure Element (SE) technologies.

Concerning security measures generally protecting Smart Grids against any potential cyber-security threats, defense strategies and some good practices need to be adopted and integrated at the AMI level, including wireless communication protocols and the architecture in use. In addition, the security of all participating components, which addresses the security of each component, including the SM, must be performed and tested. It covers device conformity, functionality, and interoperability testing.

The programmable micro-controller MCU Secure Elements used gave a Trusted Execution Environment (TEE) and Trusted Storage Environment (TSE) while all data were securely sealed at the source and successfully anchored.

Although small in size, the SEs provide security capabilities like digest functions (SHA1, SHA2, MD5, etc.) and cryptographic functions (Elliptic Curve Cryptography (ECC), Rivest-Shamir-Adleman Algorithm (RSA), Advanced Encryption Standard (AES), integrating a crypto-processor to accelerate the execution of their operations posing a maximum latency of 3 s (LoRaWAN). SEs restricted computational resources in conjunction with cloud platforms that provided services for collecting, processing, and analyzing real-time, streaming data while sealing data at the source with extremely low latency, as seen in Table 7. With specs, a SE can save up to five applications that are named codelets/applets since they are too small apps executing a particular function.

Data integrity being the property of the data used in a solution as correct, reliable, and useful for all participants, is established by combining cryptographic signatures with Blockchain technology, mainly within the secure elements. All according processes are handled by the secure elements used in all three scenarios where data of sensor measurements are connected via the gateway to the blockchain. Any external impairments, i.e., Mobile Network fallouts, for example, are compensated by the Standard Service Agreement of the platforms used. In the third implementation, for example, the MCU used with hardware crypto-support (NXP K82) uses elliptic curves (ECC, curve ed25519) as a means for asymmetric cryptography in the secure element to enhance secure data integration to the blockchain [58,59].

On the other hand, although secure elements may prove to be an overcome solution for cutting down on computational power, which is one of the most critical criteria of blockchain processing as previously mentioned, as witnessed in our research study, their effectiveness and efficiency in use require more study and consideration. Further studies could move a step further in the confirmation of blockchain technology’s security features and its implications combined with secure elements in such application environments that require a high level of identity and privacy, serving millions of decentralized nodes. The scalability issue is another operational challenge that can have an impact on blockchain adoption success for improving smart grid management and stability through smart devices, to improve real-time coordination and adjustment of the energy demand and supply.

On the outcome, in the future, we are willing to investigate effective retention and data transaction from merely the security point of view.

In particular, it would be of use in the future to research the following:

Short Term:Later a comparative analysis on the usage of virtual SE instead of a hard SE for blockchain applications, where a virtual SE is a secure sandbox that is embedded within the operating system, which creates a secure and safe environment similar to SE.

Medium Term: Application of this approach

This will include the installation of this system in many residential and small commercial settings, where more detailed tests of the ability of our SMGW to work in different neighborhood area network setups and different environments will be tested.Future tests are also proposed to improve and develop this tested open-source smart meter gateway to be integrated into a smart grid system.

Long Term

This combinational approach of SE and Blockchain can be used and adopted as the basis for P2P trading platforms. Independently of the asset that we are trading.Future research is also needed to evaluate the use of SMGW in the use cases presented below:○Smart city optimization is performed by connecting the physical object to the blockchain. When physical nodes can communicate securely within a distributed environment, it enables smart city optimization.○Tests will be contemplated to examine our SMGW capabilities to record energy use and store data related to electric vehicle charging.Rolling out the energy solution and improving these products for the new era of distributed energy that empowers utility providers to bring trust and flexibility towards local energy communities, to enable the production of renewable energy, and to prepare for the token economy to come.

## 5. Conclusions

In this paper, we have presented the hybrid approach for secure energy smart meter gateways based on blockchain and SE. As a case study, we have applied our prototype to the energy field to enable secure P2P energy trading.

We started by introducing the characteristics of both technologies, Blockchain and secure elements. Then we have described and explained how blockchain can be used as a distributed database for secure data storage and transactions. We have also mentioned that the implementation of blockchain alone is not sufficient to provide end to secure IoT platform. If data is modified before being stored on the blockchain, it will affect the consensus decision and will lead to a conflict between the validators. Since blockchain cannot penetrate lower levels, here comes the proposition of integrating a secure element. We have designed and tested three case studies using different vendors. In the first case, we have used Riddle&Code as a secure element and BigchainDB as a distributed database. However, for the second case study, we used Multos trust core as a secure element and Helium Blockchain for data storage, and for the third case, 1NCE SIM and UIBRCH as a Blockchain were implemented. The three tested implementation scenarios provide secure solution alternatives suitable for P2P (Peer to Peer) energy trading where the IoT-based devices have the role of an SMGW. The SMGW is presented as a Blockchain client that is capable of interacting with Blockchain to create an asset and register its data on the distributed database. These tested models present a secure version since we were able by the integration of a secure element to encrypt the data at the source of generation and decrypt it at the destination or the new owner.

It is a well-known fact that there is no absolute security in the IoT area and that IoT security is gradual, as the degree of IoT security is dynamic. Any high-level IoT security system can drop to a low level once an IoT hacker has discovered a security hole. Providing security measures such as Blockchain technology along with the SE is one of the ways to safeguard IoT networks from potential attacks. The main result obtained by the three experimental tests we were able to present that of a feasible, secure end-to-end system for energy trading based on these two technologies, Blockchain and SEs. The integration of a secure element at the node level (edge) provides IoT devices with moderate computational power and secures the data at the point of generation, while the use of a blockchain as a distributed database overcomes most security and trust challenges for IoT platforms.

## Figures and Tables

**Figure 1 sensors-22-09664-f001:**
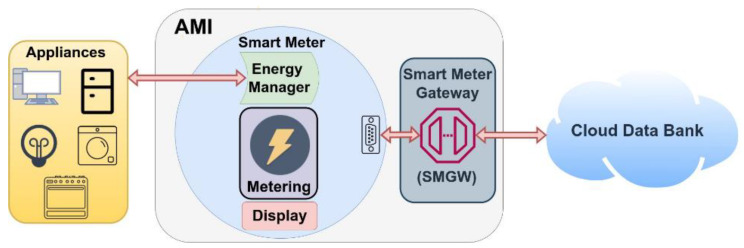
AMI Components.

**Figure 2 sensors-22-09664-f002:**
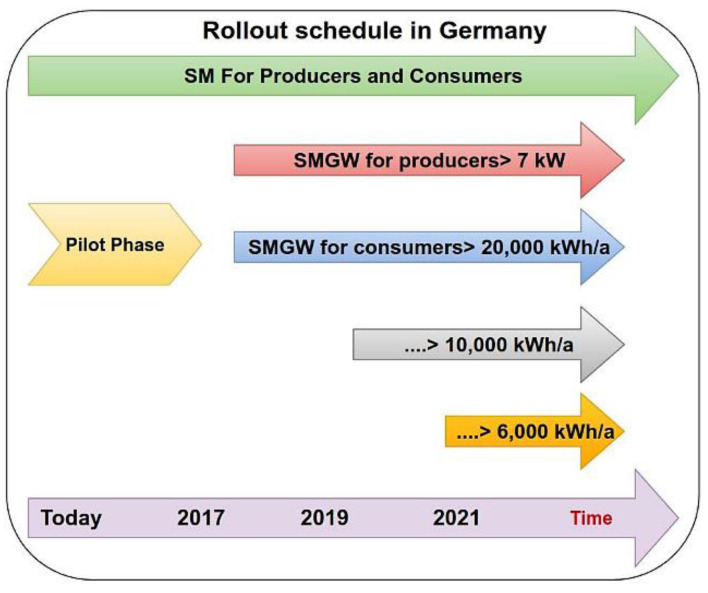
SMGW Rollout Schedule.

**Figure 3 sensors-22-09664-f003:**
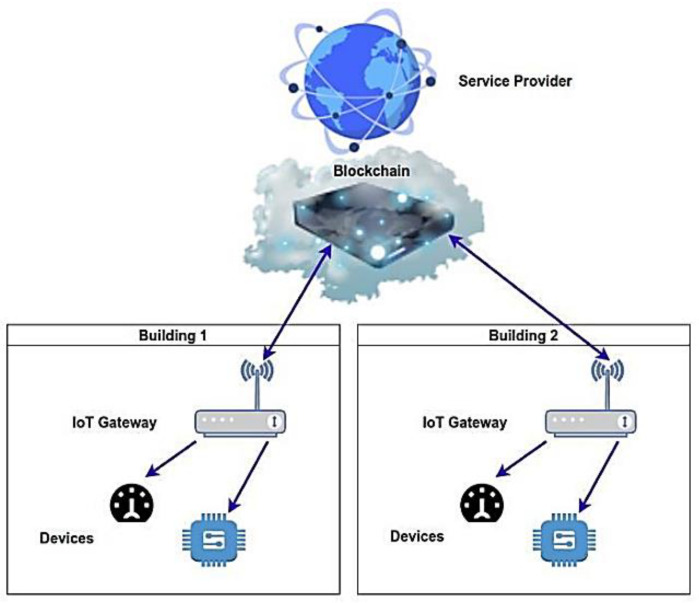
AMI-Blockchain Application Architecture.

**Figure 4 sensors-22-09664-f004:**
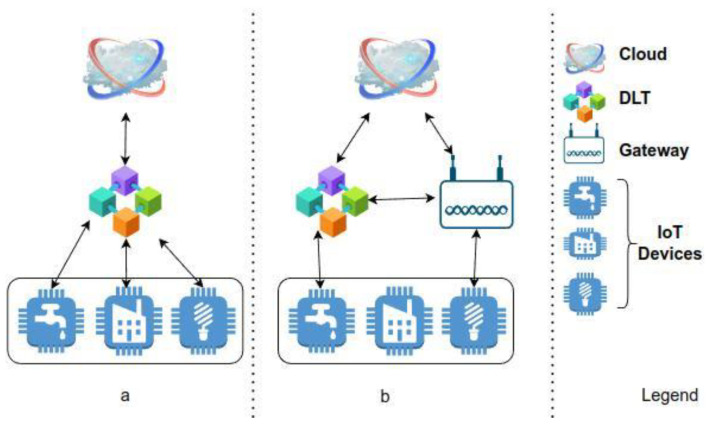
DLT and IoT Integration schemas (**a**) Without a Gateway Device, (**b**) With Gateway Device.

**Figure 5 sensors-22-09664-f005:**
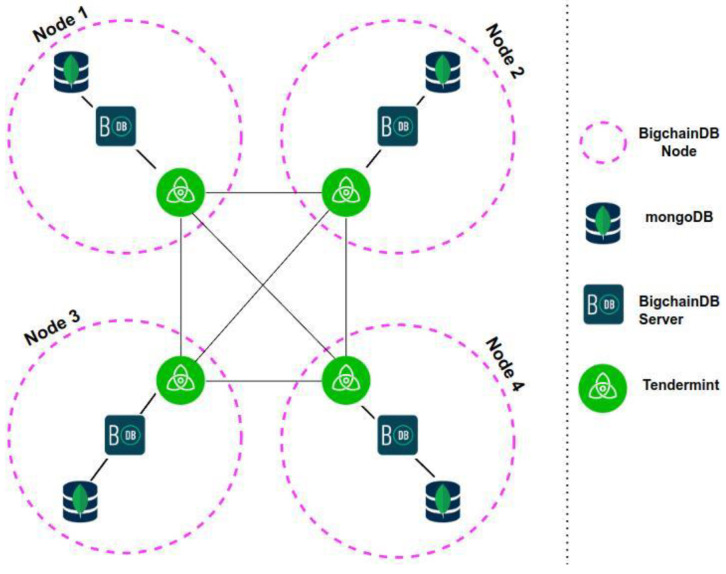
The four main components of the BigchainDB network.

**Figure 6 sensors-22-09664-f006:**
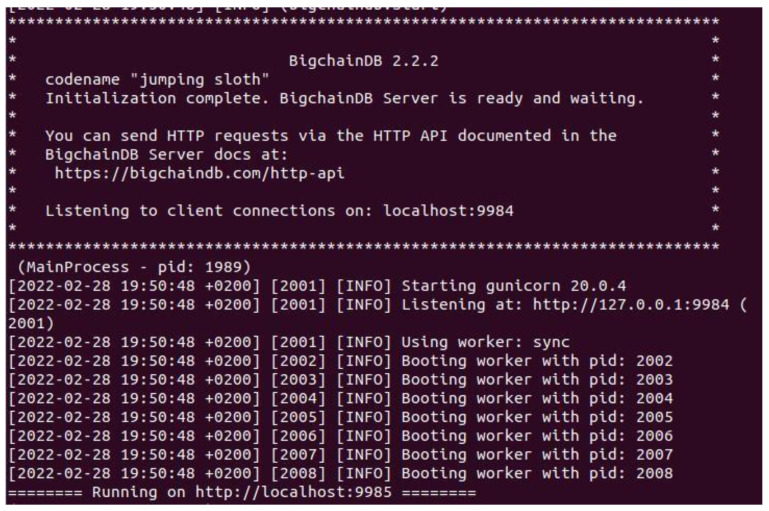
BigchainDB Server Activation.

**Figure 7 sensors-22-09664-f007:**
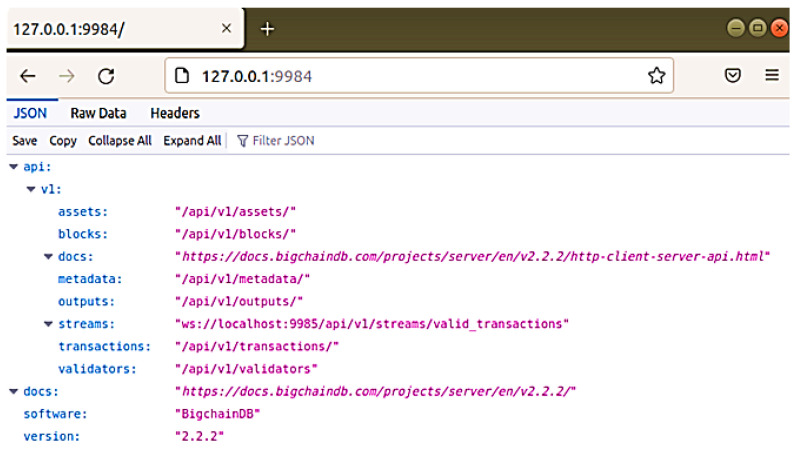
Client Connection to BigchainDB Server.

**Figure 8 sensors-22-09664-f008:**
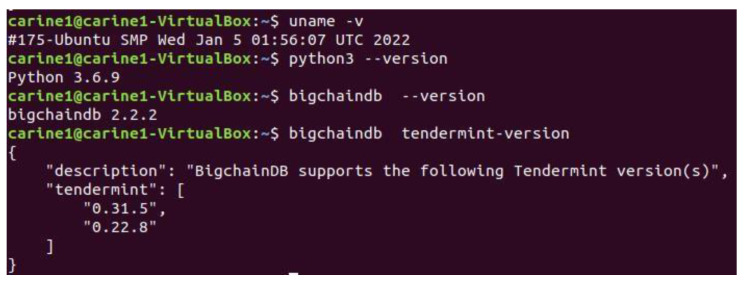
BigchainDB node environment components Versions.

**Figure 9 sensors-22-09664-f009:**
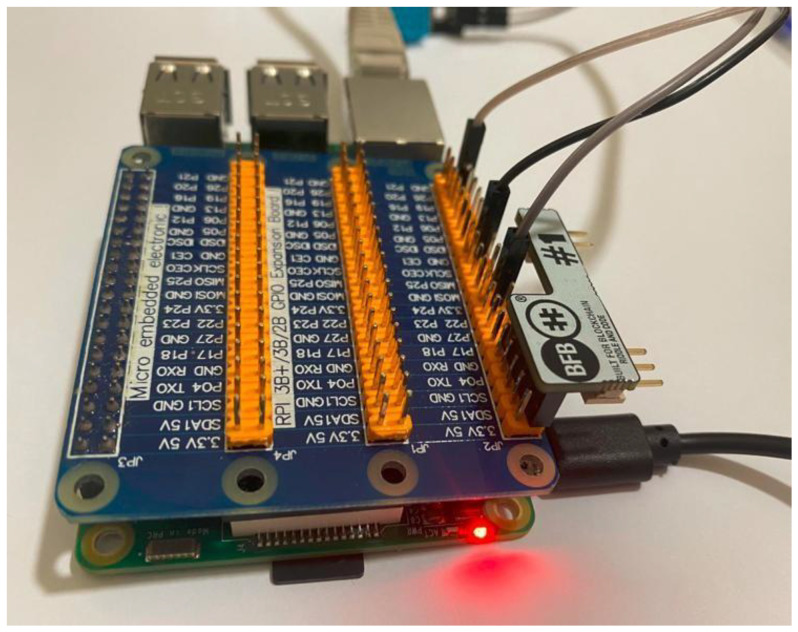
Hardware Connectivity.

**Figure 10 sensors-22-09664-f010:**
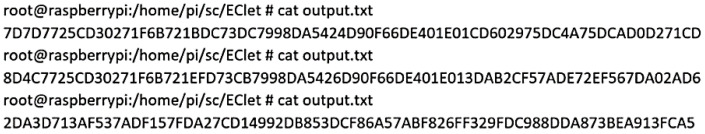
Secure Element Output.

**Figure 11 sensors-22-09664-f011:**
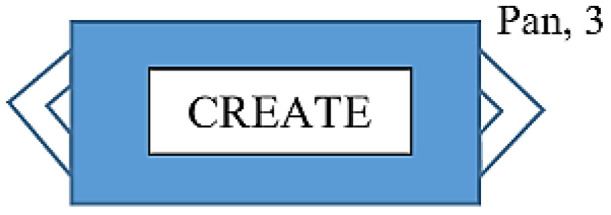
BigchainDB CREATE Transaction.

**Figure 12 sensors-22-09664-f012:**
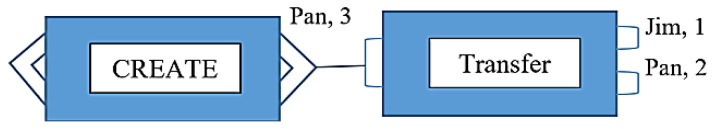
BigchainDB Transfer Transaction.

**Figure 13 sensors-22-09664-f013:**
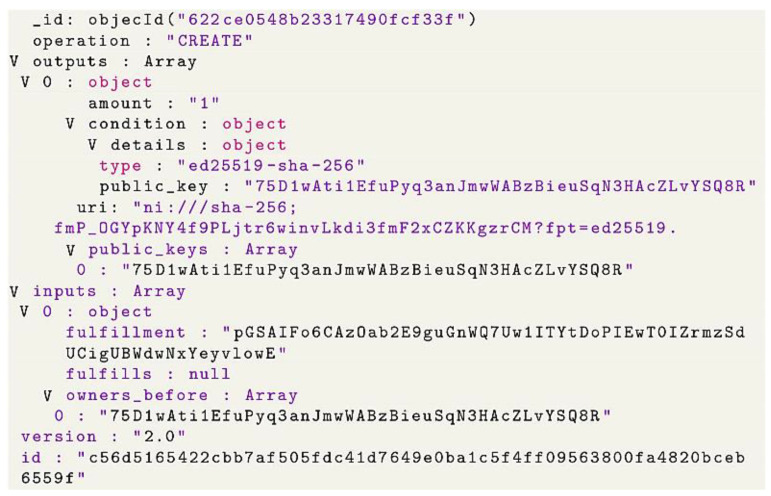
Create Transaction.

**Figure 14 sensors-22-09664-f014:**
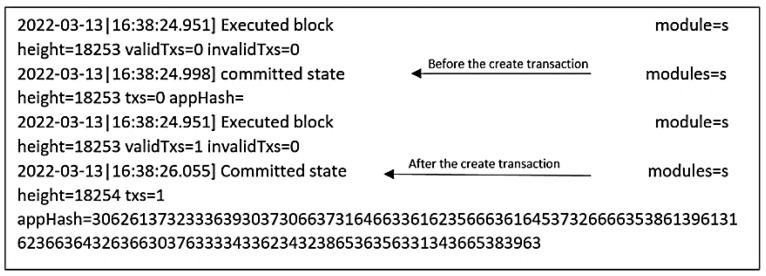
Block Creation.

**Figure 15 sensors-22-09664-f015:**
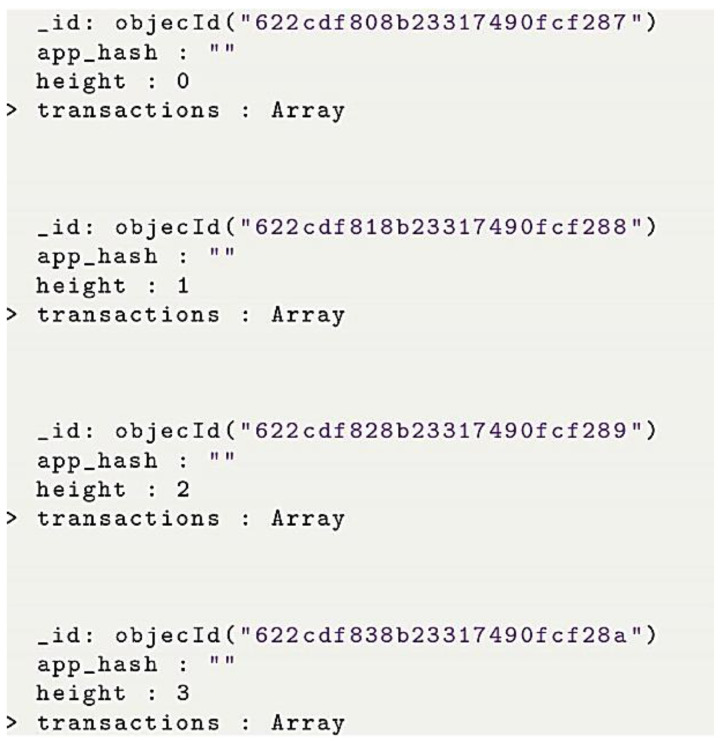
Block Enablement.

**Figure 16 sensors-22-09664-f016:**
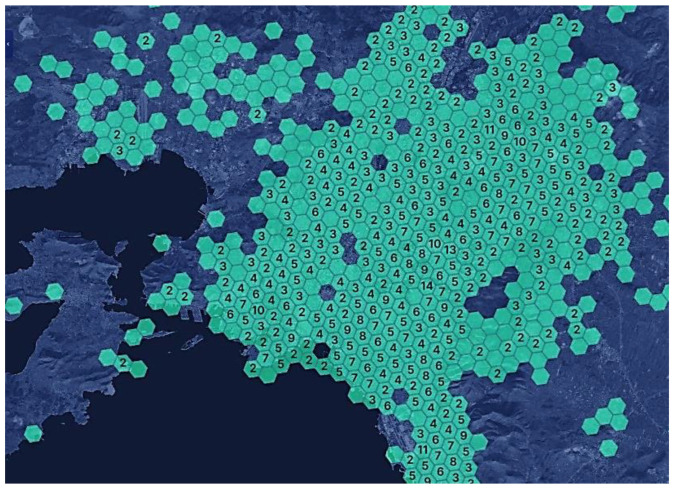
Helium Hotspots in Attica.

**Figure 17 sensors-22-09664-f017:**
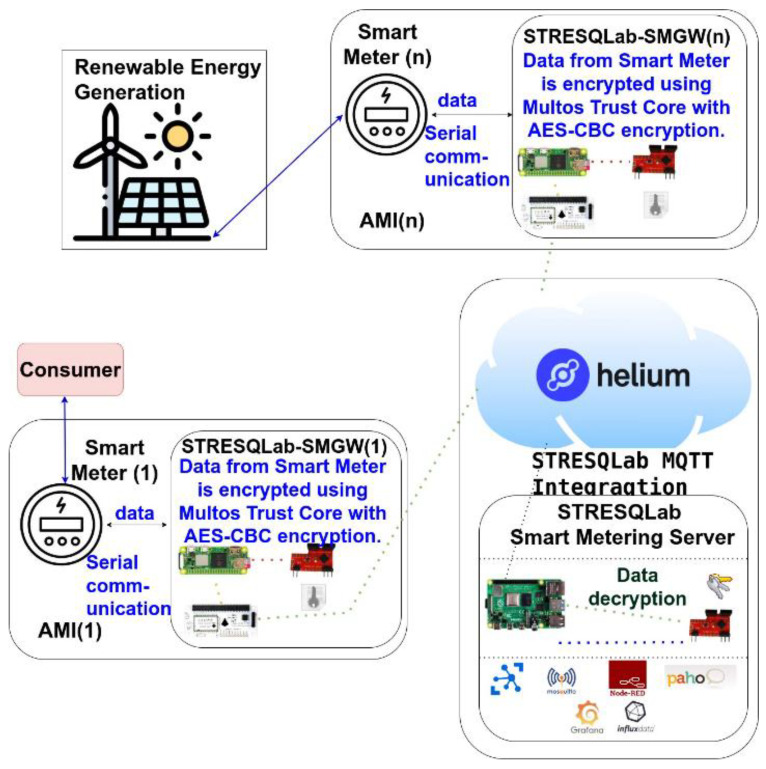
Block diagram of the IoT system.

**Figure 18 sensors-22-09664-f018:**
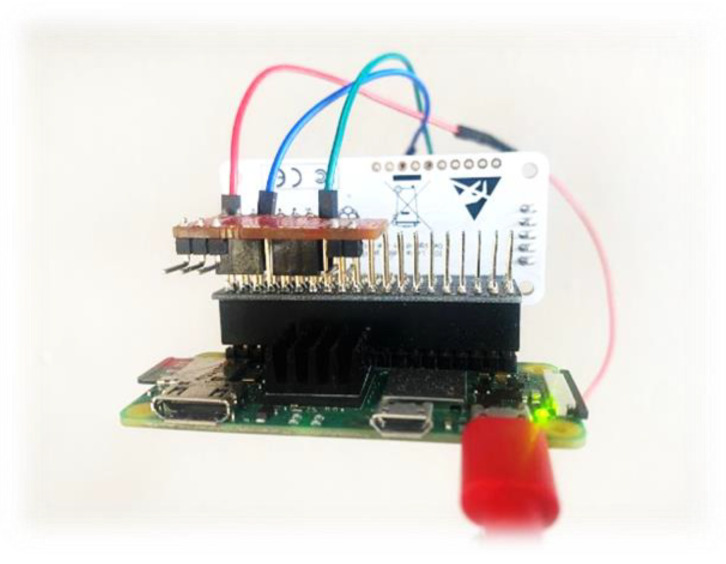
STRESQLab-SMGW.

**Figure 19 sensors-22-09664-f019:**
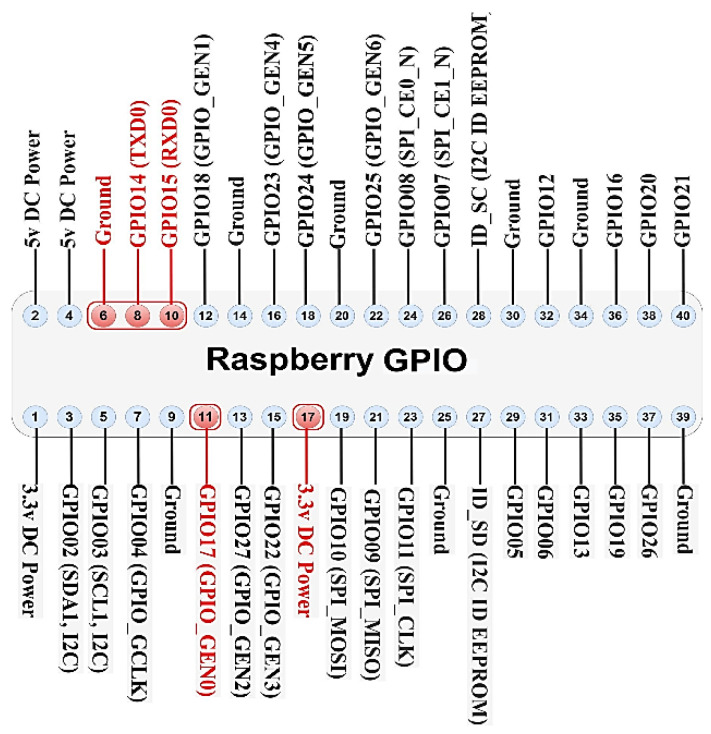
GPIO used by the module.

**Figure 20 sensors-22-09664-f020:**
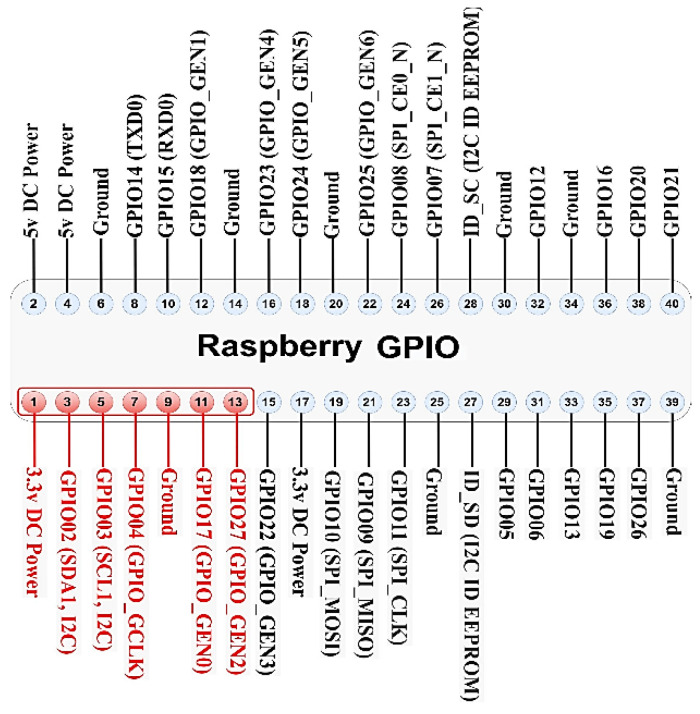
Multos Trust Core to pi GPIO.

**Figure 21 sensors-22-09664-f021:**
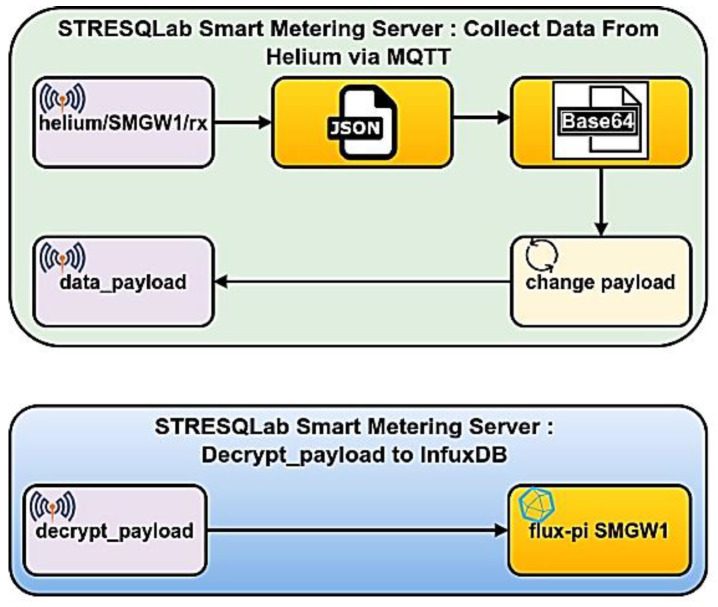
Node-Red: STRESQLab Smart Metering Server.

**Figure 22 sensors-22-09664-f022:**
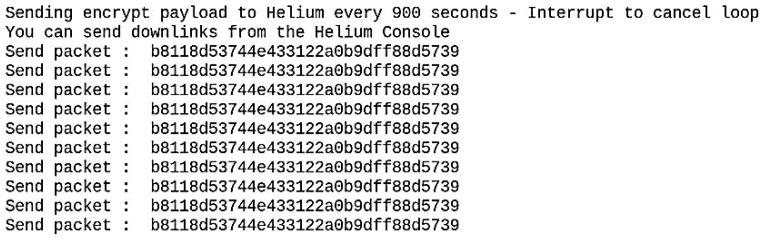
Node sending encrypted data.

**Figure 23 sensors-22-09664-f023:**
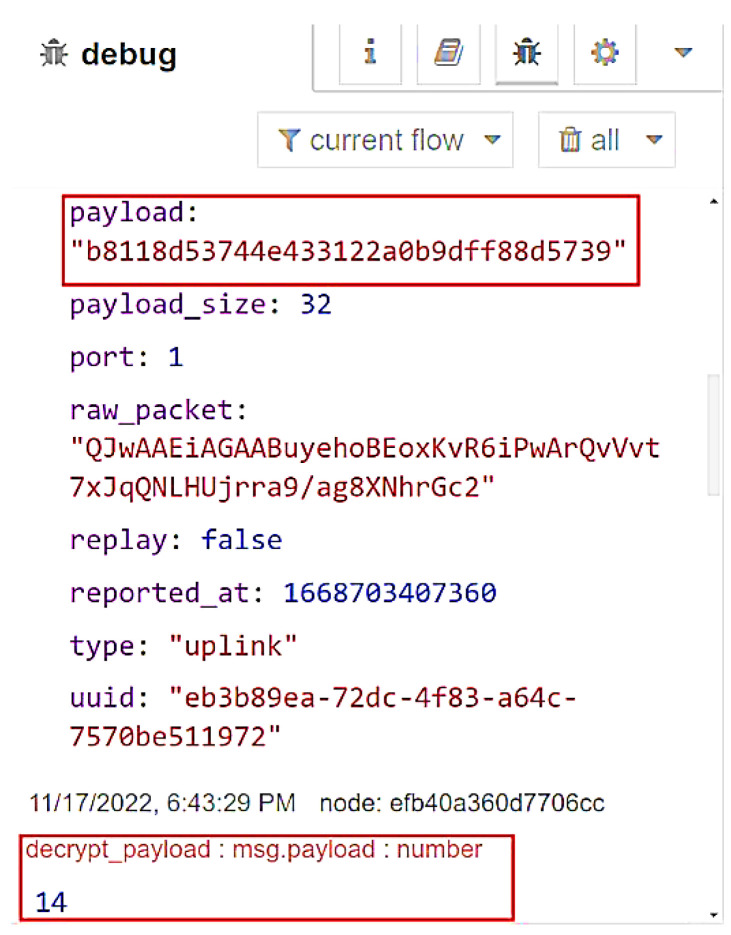
Received encrypted data.

**Figure 24 sensors-22-09664-f024:**
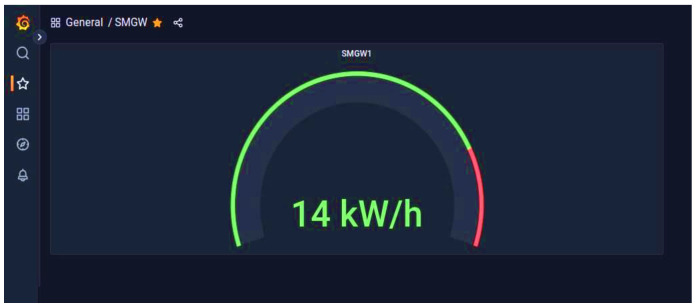
Dashboard SMGW.

**Figure 25 sensors-22-09664-f025:**
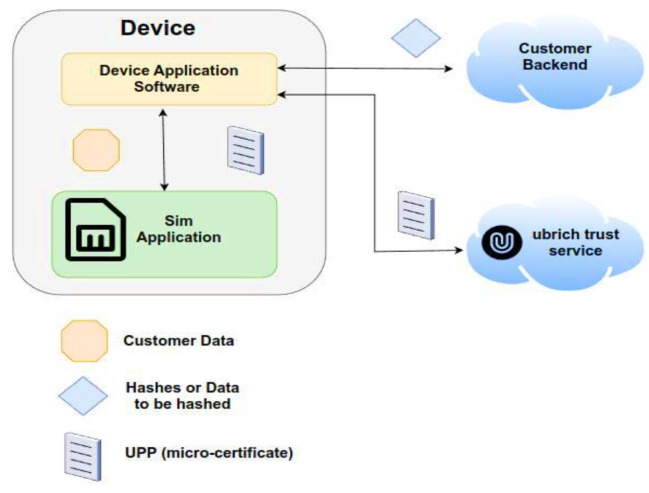
UBIRCH application structure diagram (Ubirch).

**Figure 26 sensors-22-09664-f026:**
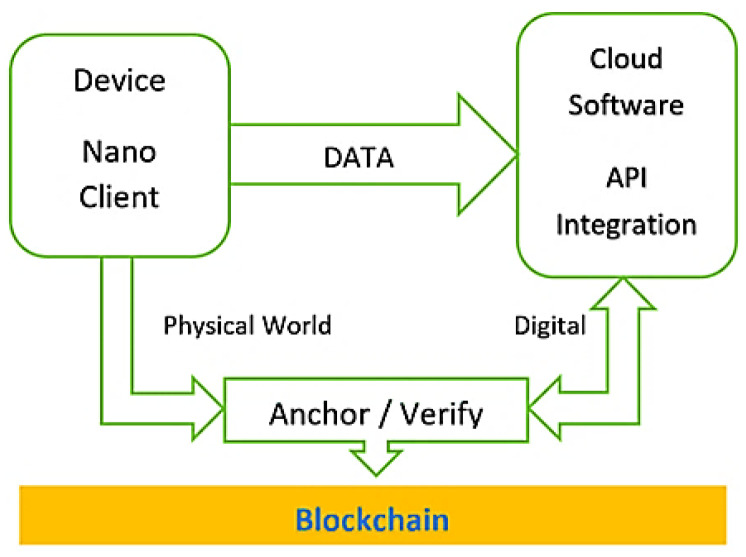
UBIRCH Chain of Trust application structure diagram (Ubirch).

**Figure 27 sensors-22-09664-f027:**
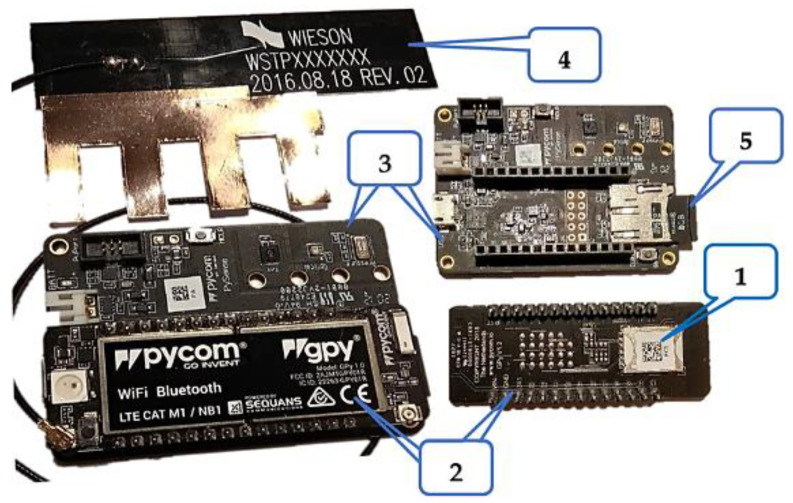
UBIRCH Testkit hardware Components.

**Figure 28 sensors-22-09664-f028:**
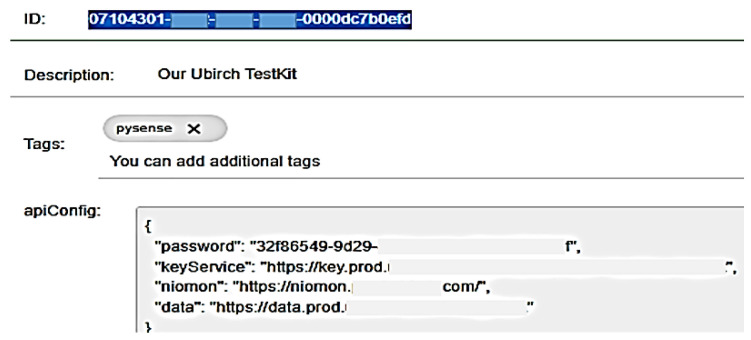
Enrolling NB-IoT device with authentication keys and corresponding backend services.

**Figure 29 sensors-22-09664-f029:**
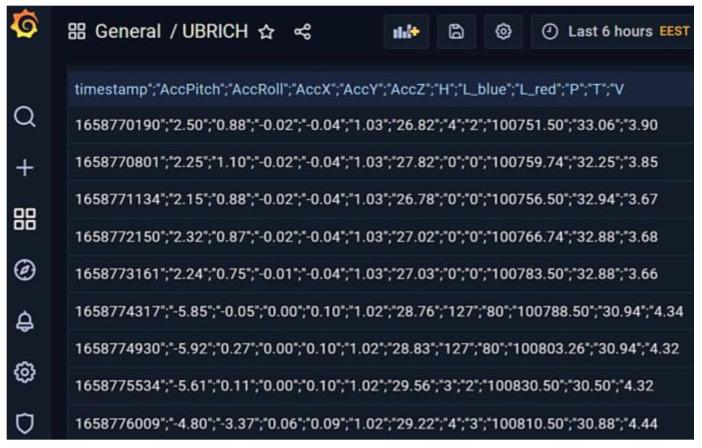
Data visualization in open-source platform Grafana.

**Figure 30 sensors-22-09664-f030:**
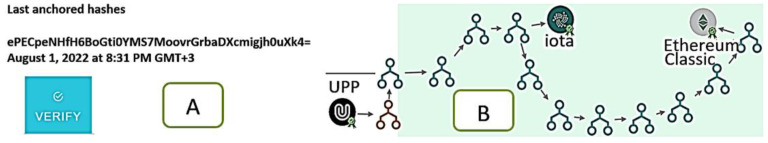
Data Verification and Blockchain Anchoring in Ethereum Classic, IOTA. (**A**) Verification button. (**B**) Blockchain procedure in graphic mode.

**Figure 31 sensors-22-09664-f031:**
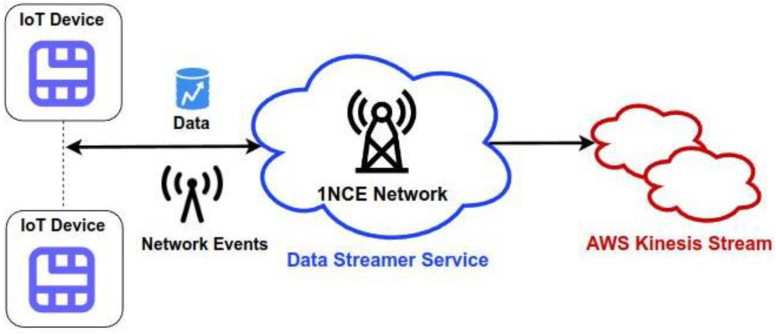
Data usage streams from 1NCE/Ubirch to AWS metrics platform.

**Table 1 sensors-22-09664-t001:** Attacks on Smart Meter devices, wireless communication technologies, and communication networks and practices to avoid them.

Advanced Metering Infrastructure	**Category**	**Attacks**	**Impact On**	**Best PRACTICES**
	Physical security	Illegitimate physical access	Access control Availability Integrity Confidentiality	Restrict the Physical access of the network devices to only specific persons Disable unused ports Use strong encryption Credential secure storage
		Data theft		
		Intentional damage		
		Firmware modification		
		Smart Meter hijacking		
		False Data Injection		
	Wireless Networks	Data theft	Integrity Access control Confidentiality Availability	Here are some measures to be taken into consideration to protect wireless networks: Use of mutual Authentication Hash function monitoring Devices secure communication. Use only approved communication channels Use of secure communication protocols MAC address filtering Access control List AES Encryption Security against masquerading parties Recently Blockchain has been adopted as a solution for a secure OTA firmware update
		Man in the middle MITM		
		Identity theft		
		Session hijacking		
		Data gathering		
		OTA firmware modification		

**Table 2 sensors-22-09664-t002:** Operating features of Helium Node.

Frequency (MHz) Code Rate Maximum Payload Size (Bytes)	Spreading Factor Bitrate (Bits/s) Latency (s)	Bandwidth (kHz) Throughput (Bits/s) Cost per 24 Bytes (€)
868	SF7	125 KHz
4/5	5470	4844.7
242	1	0.0000103

**Table 3 sensors-22-09664-t003:** Data Usage vs. time in hours.

Time Laps in Hour Max Data Transfer Rate	Number of UPPs Sent Average Data Transfer Rate	Data Used (MB) from a Total of 500 MB Peak Data Rate
30	180	3.03
1 MB/s	5.2 KB/s	13 KB/s

**Table 4 sensors-22-09664-t004:** Latency data being anchored in Blockchain and percentage of successfully transmitted Data.

Average Latency of Anchored Data in Blockchain, in Milliseconds Total Data Failed	Maximum Latency of Anchored Data in Blockchain in Milliseconds	Successfully Recorded Data in the Platform
10	65	100 %
0%		

**Table 5 sensors-22-09664-t005:** Monthly costs in Euros (€) of AWS Services/Kinetic for one SIM Card.

July 2022	August 2022	September 2022
17.12	17.1	16.55
Total costs for three months	50.88	

**Table 6 sensors-22-09664-t006:** The overheads of all three setups unveil the computing efficiency and technical characteristics and IoT communication protocols.

Case Study I: End-to-End Implementation Architecture	Case Study II: Secure Element and Helium Blockchain	Case Study III: UBIRCH Blockchain with 1NCE SIMs
Embedded System: Raspberry pi 3b+ • Processor: Broadcom BCM2837B0, Cortex-A53 (ARMv8) 64-bit SoC @ 1.4 GHz • BogoMIPS: 38.40 • Memory: 1GB LPDDR2 SDRAM. • 2.4 GHz and 5GHz IEEE 802.11.b/g/n/ac wireless LAN, Bluetooth 4.2, BLE. • Gigabit Ethernet over USB 2.0 (maximum throughput 300 Mbps)	Embedded System: Raspberry pizero2w • Processor: Broadcom BCM2710A1, quad-core 64-bit SoC (Arm Cortex-A53 @ 1GHz) • BogoMIPS: 38.40 • Memory: 512 MB LPDDR2.	Embedded System: LoPy GPy Triple–bearer MicroPython • Processor: Espressif ESP32 -Xtensa dual–core 32bit LX6 microprocessor(s), • 600 DMIPS • Memory: RAM: 520KB + 4MB External flash RAM: 8MB
Ethernet Gateway	LoRa Node: IoT LoRa Node pHAT • Chip: RAK811 LoRa^®^ Radio with full LoRaWAN^®^ Stack embedded • Communicates with the Raspberry Pi over UART only using a total of 3 GPIO Pins for the module • Low power—uses less than 50 mA During transmission • Maximum output power 100 MW (20 dBm)/adjustable from 5 to 20 dBm • High sensitivity: −148 dBm allowing extremely large range connectivity.	NB-IoT node: LTE CAT–M1/NB-IoT • Chip: CAT M1 and NB1–3GPP release 13 LTE Advanced Pro– • narrowband LTE UE-M1/NB1—Integrated baseband, RF. • Reduced TX power class option—Peak power estimations: TX current = 420 mA peak/1.5 Watt RX current = 330 mA peak/1.2 Watt • RTC: Running at 150 kHz • Security: SSL/TLS support– WPA Enterprise security Hash/encryption: SHA–MD5–DES–AES
Secure Element: RiddleCode • Crypto Accelerator Microchip 608A and Crypto Storage Microchip ATAES132A	Secure Element: Multos Trust Core • CPU: Infineon SLE78CUFX5000PHM 16 bit secure microcontroller • Free RAM: 9900 bytes	Secure Element: SIM card • MCU: NXP K82 with hardware crypto-support, • uses elliptic curves (ECC, curve ed25519)

**Table 7 sensors-22-09664-t007:** Performance comparison table of all three cases/scenarios.

Performance Facts	Case Study for Three Technological Solutions		
	I. End-to-End Implementation Architecture—Riddle&Code	II. Using Secure Element and Helium Blockchain	III. Blockchain on a SIM-1NCE/Ubirch
**Media—Communication Protocol.**	WLAN	LoRaWAN/Helium	NB-IoT
**Max Data transfer rate.**	1 MB/s	21.9 KB/s (SF8)	1 MB/s
**The number of packets sent daily.**	96	96 (max 158)	(UPPs) 96
**Average Data transfer rate.**	64 KB/s	Data Rate: 5.47 KB/s (SF7)	5.2 KB/s
**Peak Data rate**	100 KB/s	Throughput: 4.84 KB/s (SF7)	13 KB/s
**Average latency of Anchored Data in Blockchain (s)**	0.009	1	0.01
**Maximum latency of Anchored Data in Blockchain (s)**	0.04	3	0.065
**Security**	BigchainDB as a database and Riddle&Code Secure Element at the root	Helium Blockchain as Cloud Databank, Multos Trust Core at the root	Blockchain Ethereum, Ethereum Classic, and IOTAA, at the root.
**Ease of usage**	All platforms are dedicated to IoT Utilization	All platforms are dedicated to IoT Utilization	All platforms are dedicated to IoT Utilization.
**Cost per node**			
**Total Annual Blockchain services costs**	€0.01	€1.46	€7.32
**Indicative Communications cost (Flat rate)**	- (Wi-Fi LAN)	- (LoRa WAN)	€26.5 (NB-IoT)

## Data Availability

Any data presented in this study are available within the article.

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
