# Peer review of "Blockchain and Secure Element, a Hybrid Approach for Secure Energy Smart Meter Gateways"

_sensors, 2022, doi:10.3390/s22249664_

Round 1

Reviewer 1 Report

By introducing the SE, which will give IoT devices access to trustworthy computational resources, the suggested combinational strategy seeks to establish a solid root of trust in the system. The suggested method's viability is examined utilising three alternative implementation scenarios that encrypt, transfer, and store data using various Secure Element systems (SES) in conjunction with blockchain and LPWAN communication technologies. With the help of a case study for an Energy Smart Metering gateway, which enables the construction of a local Peer to Peer energy trading scheme that is end-to-end safe. The overall presentation and writing in this paper is good.

The following are a few suggestions/comments:

1.       Introduction section is short; authors may include the challenges of smart metering in smart grid systems. What motivates authors to take this problem and how blockchain is suitable for this?

2.       Literature study is well studied but authors may improve by editing some recent work on Blockchain in smart grid.

3. In Section 3 many figures need to be improved, for example figures 13, 17,20,21,23…….

4.       Computation complexity and security verification or validation are missing in the result section.

5.       Give a comparative analysis of the existing work and your proposed work, how your approach is  acceptable

6.       All the references must be unique and complete.

Reviewer 2 Report

In this paper, the authors investigate a new hybrid approach which is suitable for application to energy smart meter gateways, based on combining both blockchain and Secure Element (SE) technologies serving simultaneously the roles of a secure distributed data storage system and an essential component for building a root of trust in IoT platforms. Experiment results demonstrate the effectiveness of the proposed scheme.

However, I still have some concerns as follows.

1. In the designed architecture, how to protect the IoT gateway form potential adversaries?

2. How to ensure the consistence between the data recorded on blockchain and the collected data?

3. What are the overheads (e.g., system setup, data storage, computing efficiency, etc.) of the proposed scheme?

4. The authors have claimed the integrity of blockchain data with the static analysis of distributed storage layer. How to differentiate the authors’ work from these works which are already performed over distributed storage, such as Blockshare, VQL, and vChain+.

5. There are more opportunities for conducting meaningful experiments to comprehensively evaluate the system performance and overheads.

6. More technical papers about blockchain and IoT should be investigated and analyzed. For example:

- NutBaaS: a blockchain-as-a-service platform

- Smartphone-assisted smooth live video broadcast on wearable cameras

- A comparative study: Blockchain technology utilization benefits, challenges and functionalities

- Blockchain technology in the energy sector: A systematic review of challenges and opportunities

- Pedestrian detection with wearable cameras for the blind: A two-way perspective

- An efficient learning-based approach to multi-objective route planning in a smart city

7. What are the shortcomings of the system and future research directions? More discussion could be added.

8. Figure 21 could be improved, since the content is blurred. An algorithm flow diagram could be provided to make the proposed method more clear.

Round 2

Reviewer 2 Report

All problems have been solved.